# Unsupervised Causal Generative Understanding of Images

**Titas Anciukevičius**
Snap Inc., University of Edinburgh
`titas.anciukevicius@gmail.com`

**Patrick Fox-Roberts**
Snap Inc.

**Edward Rosten**
Snap Inc.

**Paul Henderson**
University of Glasgow

## Abstract

We present a novel framework for unsupervised object-centric 3D scene understanding that generalizes robustly to out-of-distribution images. To achieve this, we design a causal generative model reflecting the physical process by which an image is produced, when a camera captures a scene containing multiple objects. This model is trained to reconstruct multi-view images via a latent representation describing the shapes, colours and positions of the 3D objects they show. It explicitly represents object instances as separate neural radiance fields, placed into a 3D scene. We then propose an inference algorithm that can infer this latent representation given a single out-of-distribution image as input – even when it shows an unseen combination of components, unseen spatial compositions or a radically new viewpoint. We conduct extensive experiments applying our approach to test datasets that have zero probability under the training distribution. These show that it accurately reconstructs a scene's geometry, segments objects and infers their positions, despite not receiving any supervision. Our approach significantly out-performs baselines that do not capture the true causal image generation process.

## 1 Introduction

Most machine learning approaches make the assumption that at test time, they are applied to data drawn from the same distribution as seen during training [14, 91, 73, 21, 33]. This means the generalization guarantees of statistical learning theory apply [127]. However, this does not apply to images drawn from a different distribution – recent works have shown this for images taken from unfamiliar viewpoints [4, 1, 8], shifted by few pixels [7], and showing scenes with an unseen composition of objects [11, 110, 114, 36, 37, 26, 70]. It has been suggested that this is because they learn spurious *shortcuts* [36] to achieve low training loss, but which do not capture the true *causal* relationship. However, when deployed, machine learning methods often encounter observations drawn from a previously unseen distribution.

In this work, we consider the task of transforming a single observed image into a detailed representation of the scene it depicts, providing explicit information about its 3D structure such as object locations, shapes and appearances. We focus on the challenging setting where at test time, we see images depicting scenes that have zero probability in the training distribution (Fig. 1). We adopt an *unsupervised* approach to learning, avoiding the need for manual annotation of object masks, 3D positions, and similar – we require only a dataset of posed multi-view images.

Our approach is to build a generative model (Sec. 3) jointly over image pixels and the 3D world they depict, that can be robustly inverted to infer the latent factors that gave rise to an input image [116, 129, 22]. If such a model is to support generalization to out-of-distribution (OOD) data, it should reflect the underlying causal model of the environment from which the data arose [102, 116].

36th Conference on Neural Information Processing Systems (NeurIPS 2022).

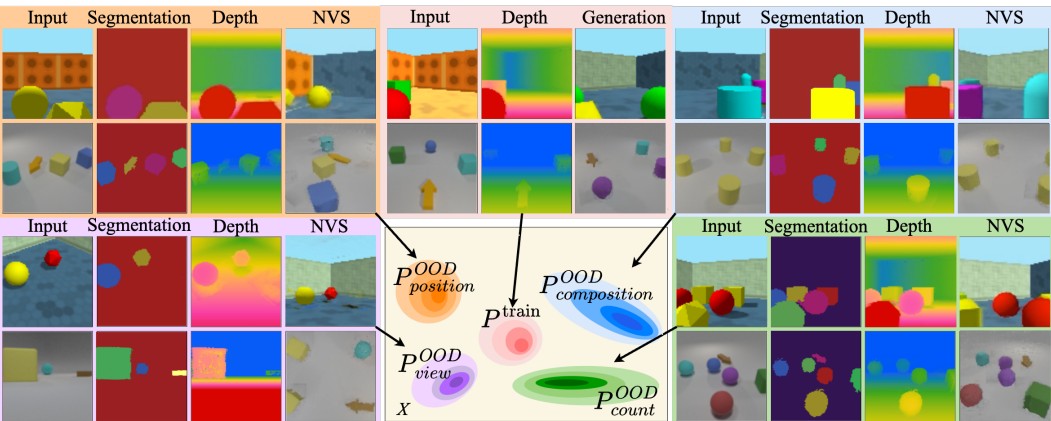

Figure 1: Our causal generative model learns to represent images from one probability distribution $P^{train}$ but also allows interventions on the model to represent a set of different distributions. This allows both (i) inference of object segmentations, depth-maps, and novel views (NVS) for input images drawn from different distributions (e.g. taken from radically different camera viewpoints $P^{OOD}_{views}$, containing unseen compositions of objects $P^{OOD}_{composition}$, or unseen number of objects $P^{OOD}_{count}$); (ii) generating plausible 3D scenes containing multiple objects ('Generation', top center).

In practice this means the conditional distributions in the model should correspond (as far as possible) to *independent mechanisms* [113, 56] in the world. Thus, when mechanisms or physical processes in the environment change, we expect much of our model to remain applicable; this has been called *sparse mechanism shift* [116]. For example, even if objects are no longer placed according to the same patterns as at training time, we still expect the appearances of object types to remain consistent.

In general it is impossible to recover such a causal model purely from observational data [102, 116]. We therefore embed in our model the physical knowledge that the world is composed of 3D objects of different shapes, which may appear at different locations, imaged by a camera subject to the laws of 3D geometry and perspective projection. Each object has *separate* appearance and position representations, ensuring disentanglement, so the same object can be represented invariantly in different locations. The final image is synthesized by volumetric rendering of the latent 3D shapes placed in a 3D scene space according to their latent positions. This structure means the model is *compositional*: having learnt about one type of object in one context, it can also represent and perform inference about a similar object in different contexts.

We perform inference on OOD data by *intervening* on our model [102, 105] – i.e. replacing certain conditional distributions (or mechanisms) that are no longer appropriate and then doing posterior inference. To ensure that the inference method itself supports OOD generalization we cannot employ amortized variational inference [65, 109], as an encoder network is in general not robust to changes of distribution [89, 36]. Instead, we develop a novel Markov chain Monte-Carlo (MCMC) inference scheme, that finds posterior samples for a given test image without any encoder network.

Targeting OOD generalization requires important design choices in our model, that differ from other generative models of 3D scenes [97, 122, 72]. First, we use a non-learnt rendering mechanism, as this is guaranteed to generalise correctly to OOD data (e.g. novel viewpoints); it contrasts with prior works that learn the rendering process (e.g. with a CNN mapping rendered features to pixels [97, 94]). Second, we use an explicit disentangled representation of the assumed underlying causal variables (e.g. object shapes and positions), which allows performing interventions and counterfactual inference; this contrasts with works that model the scene without disentangling the proper causal variables, such as spatial mixture models [33, 122]. Third, we separate the mechanisms (conditionals) for per-object appearances and for scene layouts, so the latter can be intervened on without affecting the former; this contrasts with methods that have a single global decoder [72, 94].

To evaluate our approach (Sec. 4) we create challenging test datasets that have zero probability under the corresponding training distribution, yet share aspects of its structure. We show that our model can generalize to unseen numbers of objects, unseen compositions, and radically new camera viewpoints – all significantly better than existing works. It successfully localizes objects in the 3D scene, reconstructs their 3D shapes, and hence predicts depth-maps and instance segmentation masks.

To summarise, our core contribution is **the first unsupervised framework for inference of explicit object-centric 3D scene representations, that generalizes to out-of-distribution scenes**. Our secondary contributions are: (i) experimental evidence that current methods do not generalize to images with a radically new camera viewpoint, a different number of objects, or an unseen composition of objects; (ii) a novel generative model of 3D scenes based on multi-object radiance fields with explicit object positions and volumetric rendering; and (iii) a novel MCMC inference scheme exploiting the structure of our model, that allows inferring 3D scenes from a single OOD image.

## 2   Related Work

Recently numerous works have observed that modern learning-based computer vision methods fail to generalise on out-of-distribution data [4, 1, 46, 48, 51, 71, 7, 11, 110, 142, 114, 36, 37, 92, 87, 10, 123, 26, 141, 76] and constructed various benchmarks [68, 47, 8, 42, 62, 136, 131]. Others have tried to improve OOD generalisation, the most relevant by aiming to learn relations that are invariant across training domains [6, 119, 3, 2, 100, 74, 118, 104] or that maximize sparsity of interactions [39, 38]. These typically assume that multiple differently-distributed datasets are available during training, and mainly address the supervised setting (e.g. image classification). Like the latter, our focus is on enabling OOD generalisation, but in contrast to them, we address unsupervised object-centric generative modelling. Moreover, we do not rely on multiple *training* datasets to discover causal factors, but instead directly incorporate universal knowledge such as separation of the world into objects.

Our work is also connected to the *vision as inverse graphics* paradigm [41, 9, 63, 140, 27, 111, 90]. In this setting, it is assumed that we have access to (maybe parametric) 3D models of objects, and wish to find suitable pose and other parameters to explain an input image [83, 75, 53, 111, 112, 52]. Like our work, these typically use a test-time optimisation; unlike ours, they do not attempt to learn priors on object layout nor shapes – instead assuming these are known *a priori*.

Neural implicit scene representations [124] learn a continuous representation of a 3D scene from 2D images using neural rendering, either by explicit volumetric rendering [88, 85, 132, 135, 98] or with CNN post-processing of rendered features [121, 120]. These initial works fitted individual scenes without learning common characteristics, therefore requiring many images as input. This was addressed by sharing models across different scenes [138, 125, 77, 35, 103, 96, 55], allowing inference of novel views from one or few images. All these methods model a scene as a single entity without decomposing it into individual objects, meaning manipulating the scene (e.g. moving single objects) is not possible. In contrast, [99] divides a scene into objects, but requires detailed manual annotations to do so, and does not support inference from few images; [28] relies on weaker supervision in the form of ground-truth object masks. [139, 122] discover such a decomposition automatically (though with depth supervision for [122]), however they rely on spatial mixtures to assign each point in space to an object, without any explicit, controllable representation of object positions. This entanglement of latent position and appearance means they are not guaranteed to generalise to OOD combinations of position and appearance [81]. Finally, [43] supports composition of multi-object scenes using neural scattering functions – but these must be learnt from multiple views of single objects (a form of weak supervision), with no probabilistic model over appearances nor layouts.

Other approaches extend neural rendering to the generative setting [115, 97, 94, 95, 72, 19, 25, 24], allowing sampling objects or scenes *a priori*. However, these do not allow us to perform inference in the OOD setting, as they have components that do not reflect the causal, compositional structure of the world (e.g. a monolithic latent space lacking object-centric representations, or a learnt neural renderer that will not generalise to OOD viewpoints nor compositions). In contrast, there are object-centric generative models that can sample plausible images and perform inference – but only in 2D, without reasoning over a latent 3D scene representation, and therefore without supporting 3D tasks such as depth prediction. Some use a full-image spatial mixture model [33, 93, 67, 31, 32, 60] or alpha stacking [126, 128]; others model images as composed of smaller patches or sprites [34, 57, 5], learning both the distribution of appearances and compositions; others use compositional energy-based models [29, 80]. A 3D extension of these latter is proposed by [45], but requires videos for inference at test time and does not support OOD generalisation. Other methods take purely discriminative approaches to unsupervised segmentation [82, 61, 66]. In the supplementary, we provide a table comparing the capabilities of our method to closely related works.

# 3 Method

Our goal is to infer an explicit object-centric representation of a 3D scene from one or multiple images, even when they show scenes lying outside the distribution observed during training. We achieve this by learning a compositional generative causal model jointly over multi-view images and the scenes they depict, as described in Sec. 3.1. For training (Sec. 3.2) we use only posed multi-view images $(\mathbf{x}_1, \mathbf{v}_1), ..., (\mathbf{x}_N, \mathbf{v}_N)$ drawn from a training distribution $P^{train}(\mathbf{x}, \mathbf{v})$ without any annotations such as depth-maps, bounding-boxes or segmentation masks. At test time, input images are drawn from a distribution $P^{OOD}(\mathbf{x}, \mathbf{v})$ with support disjoint from the training distribution. Performing inference over the generative model using our proposed framework (Sec. 3.3) yields an explicit object-centric representation, including 3D object shapes, positions and appearances.

## 3.1 Compositional Generative Causal Model

We model a multi-view set of $N$ images $\{\mathbf{x}_1 \ldots \mathbf{x}_N\}$ as caused by a single 3D scene $\mathcal{S}$ being rendered from viewpoints $\{\mathbf{v}_1 \ldots \mathbf{v}_N\}$, by a function $C(\mathcal{S}, \mathbf{v})$. We now describe this scene representation, then the generative process by which it is sampled.

**Scene representation.** The scene $\mathcal{S} = \left( \mathbf{s}_{bg}, \{(\mathbf{s}_i^{app}, \mathbf{s}_i^{pos})\}_{i=1}^O \right)$ is composed of a 3D background component $\mathbf{s}_{bg}$ describing the background's shape and color, and 3D objects indexed $i = 1 \ldots O$ with shape and color described by $\mathbf{s}_i^{app}$ and explicit 3D positions $\mathbf{s}_i^{pos}$. Each $\mathbf{s}_i^{app}$ explicitly represents the 3D appearance of an object as a neural radiance field (NeRF) [88] in a *canonical space* (e.g. with the object centered at the origin). The positions $\mathbf{s}_i^{pos}$ specify objects placement in the global 3D *scene space*. While most prior work parametrizes 3D object positions as coordinates, we represent them by 1-hot vectors choosing from a set of plausible candidate locations to use as the center position of the object. As we explain in Sec. 3.3, this makes gradient-based optimization easier.

**Generative process for $\mathcal{S}$.** We first sample a high-level latent scene embedding $\mathbf{z}^g \sim \mathcal{N}(\mathbf{0}, \mathbf{I})$, that will model correlations between objects and learn the typical composition of a scene [57, 5]. The individual object appearances are specified by Gaussian latent variables $\mathbf{z}_i^{shape}$ and $\mathbf{z}_i^{col}$ that respectively encode the shape and color of the $i^{\text{th}}$ object; they are conditioned on $\mathbf{z}^g$, with mean and log-variance given by a fully-connected network $\zeta_\theta(\mathbf{z}^g)$ with weights $\theta$. [1] The position $\mathbf{z}_i^{pos}$ is specified by a categorical variable, with logits given by $\xi_\theta(\mathbf{z}^g)$. We similarly introduce latents $\mathbf{z}_{bg}^{shape}$ and $\mathbf{z}_{bg}^{col}$ to encode the shape and color of the background. For brevity, we will write $\mathbf{z}^s = \{\mathbf{z}_{bg}^{shape}, \mathbf{z}_{bg}^{col}, \mathbf{z}_{1...O}^{shape}, \mathbf{z}_{1...O}^{col}, \mathbf{z}_{1...O}^{pos}\}$. The latents $\mathbf{z}^s$ are mapped to the scene $\mathcal{S}$ by a function $S_\theta$. This sets $\mathbf{s}_i^{pos}$ equal to $\mathbf{z}_i^{pos}$, and derives the object NeRF representations $\mathbf{s}_i^{app}$ from $\mathbf{z}_i^{shape}$ and $\mathbf{z}_i^{col}$ as described in the next paragraph. The probability of an image $\mathbf{x}_n$ given its camera viewpoint $\mathbf{v}_n$ is then

$$p_\theta(\mathbf{x}_n \,|\, \mathbf{v}_n) = \iint f_\mathcal{N}(\mathbf{x}_n; \, C(S_\theta(\mathbf{z}^s), \, \mathbf{v}_n), \, \sigma^2) \, p_\theta(\mathbf{z}^s \,|\, \mathbf{z}^g) \, p_\theta(\mathbf{z}^g) \, \mathrm{d}\mathbf{z}^s \mathrm{d}\mathbf{z}^g \tag{1}$$

where $C(\mathcal{S}, \mathbf{v})$ renders the scene described by $\mathcal{S}$ from viewpoint $\mathbf{v}$, and $f_\mathcal{N}$ represents a factored Gaussian likelihood over the $H \times W \times 3$ pixels of the image, with fixed standard deviation $\sigma$. The probability of a composition $\mathbf{z}^s$ of objects and background in a scene is given by

$$p_\theta(\mathbf{z}^s \,|\, \mathbf{z}^g) = p_\theta(\mathbf{z}_{bg}^{shape} \,|\, \mathbf{z}^g) p_\theta(\mathbf{z}_{bg}^{col} \,|\, \mathbf{z}^g) \prod_{i=1}^O p_\theta(\mathbf{z}_i^{shape} \,|\, \mathbf{z}^g) \, p_\theta(\mathbf{z}_i^{col} \,|\, \mathbf{z}^g) \, p_\theta(\mathbf{z}_i^{pos} \,|\, \mathbf{z}^g) \tag{2}$$

We emphasise $p_\theta(\cdot)$ models different distributions for each object and variable; it does not assume the different scene variables constituting $\mathbf{z}^s$ are I.I.D. Hence, $p_\theta(\mathbf{z}^s \,|\, \mathbf{z}^g)$ can model any relationship among object locations, shapes and colors, which is necessary to sample scenes with plausible relationships among scene components. The graphical model is illustrated in supplementary Fig. 9 in Sec. 7.

**Rendering the scene $\mathcal{S}$.** The rendering process $C(\mathcal{S}, \mathbf{v})$ outputs an image $\mathbf{x}$ for a camera viewpoint $\mathbf{v}$, given our explicit compositional representation of a scene $\mathcal{S}$. Recall $\mathcal{S}$ contains a 3D background component $\mathbf{s}_{bg}$ and a set of object components $\{\mathbf{s}_i^{app}, \mathbf{s}_i^{pos}\}_{i=1}^O$; for brevity we will identify the

---

[1]See the supplementary for all network architectures

background as component $i = 0$, with $\mathbf{s}_0^{pos}$ fixed to the origin. We extend multi-component neural radiance fields (NeRF) [88, 85] to support explicit placement of objects in the 3D scene according to the position variables $\mathbf{s}_i^{pos}$. Specifically, the latent codes $\mathbf{z}_i^{shape}$ and $\mathbf{z}_i^{col}$ for the $i^{\text{th}}$ object parametrize a learnt function $f_\theta^*(\mathbf{q}^*; \mathbf{z}_i^{shape}, \mathbf{z}_i^{col})$, that maps points $\mathbf{q}^*$ in the canonical space of the object to a color $c \in [0, 1]^3$ and density $\sigma \in \mathbb{R}^+$. We place each object at its 3D position $\mathbf{s}_i^{pos}$ by convolving its density and color functions with a one-hot location indicator:

$$f_i(\mathbf{q}) = \iiint_{\mathbf{q}^*} s_i^{pos}(\mathbf{q}^*) \cdot f_\theta^*(\mathbf{q} - \mathbf{q}^*; \mathbf{z}_i^{shape}, \mathbf{z}_i^{col}) \, d\mathbf{q}^* \equiv (c_i(\mathbf{q}), \sigma_i(\mathbf{q})) \tag{3}$$

where $\mathbf{q}$ is a position in scene space, and $s_i^{pos}(\mathbf{q})$ is an indicator function with a unit impulse if point $\mathbf{q}$ is chosen as the object center position by the 1-hot indicator $\mathbf{s}_i^{pos}$. As in [139] we divide the scene space into foreground/background regions and only render the corresponding components in each.

Given the placed object densities $\sigma_i$ and colors $c_i$, we calculate the color of each pixel in the image $\mathbf{x}$ by casting a ray $\mathbf{r}(t) = \mathbf{x_0} + t\mathbf{d} \in \mathbb{R}^3$ from the pixel in direction $\mathbf{d}$ through a camera at position $\mathbf{x_0}$, summing the contributions from different objects [85, 86]:

$$C(\mathcal{S}, \mathbf{v})[\mathbf{r}] = \int_0^\infty T(t) \sum_{i=0}^O \sigma_i(\mathbf{r}(t)) \cdot \mathbf{c}_i(\mathbf{r}(t)) dt, \quad \text{where} \quad T(t) = \exp\left(-\int_0^t \sum_{i=0}^O \sigma_i(\mathbf{r}(t')) dt'\right) \tag{4}$$

**Continuous relaxation of object placement.** To allow gradient-based training and inference, our generative process must be differentiable. We therefore relax the categorical position variable to a Gumbel-Softmax [54, 84]. This approach ensures we always receive non-zero gradients of the image with respect to every possible object position, easing optimisation. This is in contrast to models based on spatial transformers [97, 134, 78], which can get stuck in local minima if the model has a poor initial prediction, as the gradient of pixels wrt position is zero if the predicted and true positions do not overlap. Note that for a discretized representation (e.g. voxels [79] or triplanes [18]), the object placement operation Eq. 3 can be efficiently implemented as a convolution operation in the Fourier domain – exploiting the Fourier transform's *sifting property* [16, 5].

## 3.2 Training

We train our generative model from a dataset of images containing $K$ views for each of $T$ scenes. The model includes three learnable components, with parameters $\theta$: (i) $f_\theta^*(\mathbf{q}; \mathbf{z}^{shape}, \mathbf{z}^{col})$ that represents a 3D object as a function from position to color and density conditioned on the object appearance embedding; (ii) $f_\theta^{bg}(\mathbf{q}; \mathbf{z}_{bg}^{shape}, \mathbf{z}_{bg}^{color})$ that similarly represents the 3D background; (iii) $\zeta_\theta$ and $\xi_\theta$, that map the global scene latent $\mathbf{z}^g$ to parameters of the object and background latents $\mathbf{z}^s$. We train the model using autoencoding variational Bayes [65, 109]. The posteriors over Gaussian latent variables are all diagonal Gaussians (parametrized by mean, and log-variance for $\mathbf{z}^g$), whilst for positions the posterior is Gumbel-Softmax (parametrized by logits). We use two encoder networks to parametrize these variational posteriors. $\text{enc}_\phi^s(\{\mathbf{x}_n, \mathbf{v}_n\}_{n=1}^M)$ parametrizes $q(\mathbf{z}^s | \{\mathbf{x}_n, \mathbf{v}_n\}_{n=1}^M)$; for efficiency, we pass it only a subset of $M < K$ images. It encodes each observed image and its viewpoint $(\mathbf{x}_n, \mathbf{v}_n)$ independently then sums the results (as in [35]) before outputting the posterior parameters; this ensures the encoder is invariant to the ordering of images. $\text{enc}_\phi^g(\mathbf{z}^s)$ parametrizes $q(\mathbf{z}^g | \mathbf{z}^s)$, and takes the lower-level latent code $\mathbf{z}^s$ as input.

For stable training, we adopt a two-stage approach. We first train the model to reconstruct $\mathbf{x}_{1...K}$, via the object-level latent space $\mathbf{z}^s$, ignoring the scene-level latent $\mathbf{z}^g$, i.e. maximizing the following loss:

$$\mathcal{L}^s = \mathbb{E}_{q_\phi(\mathbf{z}^s | \{\mathbf{x}_n, \mathbf{v}_n\}_{n=1}^M)} \left[ \sum_{n=1}^K \log f_\mathcal{N}(\mathbf{x}_n; C(S_\theta(\mathbf{z}^s), \mathbf{v}_n), \sigma^2) \right] \tag{5}$$

After this has converged, we learn the scene-level latent space by maximizing

$$\mathcal{L}^g = \mathbb{E}_{q_\phi(\mathbf{z}^g | \mathbf{z}^s)} \left[ \mathbb{E}_{q_\phi(\mathbf{z}^s | \{\mathbf{x}_n, \mathbf{v}_n\}_{n=1}^M)} \log p_\theta(\mathbf{z}^s | \mathbf{z}^g) \right] - D_{\text{KL}} \left[ q_\phi(\mathbf{z}^g | \mathbf{z}^s) \| p_\theta(\mathbf{z}^g) \right] \tag{6}$$

We use Adam for optimization [64], $\beta$-weighting of KL terms [50], and approximate each of the above expectations by a single sample. We also further approximate $\mathcal{L}^s$ by rendering only a random subset of pixels per minibatch. More implementation details are in the supplementary material.

### 3.3 Inference for out-of-distribution (OOD) images

At test time, we assume images are sampled from a distribution disjoint from the training distribution. This means that directly performing posterior inference under our model (which has learnt the training distribution, and ideally assigns zero probability to OOD test images) is not sound. We therefore make appropriate *interventions* on our model [102], taking advantage of its causal nature. For example, when the distribution of object arrangements is different at test time, we replace the learnt prior $p_\theta(\mathbf{z}^s)$ on object arrangements with an uninformative uniform prior. [2] Moreover, the variational encoder networks $\text{enc}_\phi^s$ and $\text{enc}_\phi^g$ used during training are not suitable for use at test time, due to domain shift in their inputs. Therefore, our framework instead directly samples the posterior distribution of latent variables given an observed image, using Markov chain Monte-Carlo (MCMC) inference [69].[3]

Our novel MCMC scheme alternates Langevin dynamics (LD) [12, 130] and Metropolis-Hastings (MH) [44] steps, to infer the latent scene variables $(\mathbf{z}^s, \mathbf{z}^g)$ from a *single* observed image $\mathbf{x}^*$ with viewpoint $\mathbf{v}^*$. The MH steps encourage the Markov chain to make large jumps between modes of the posterior, while the LD steps generate high-probability samples with less exploration. Each LD step ascends the gradient of

$$\log f_\mathcal{N}(\mathbf{x}^*; C(S_\theta(\mathbf{z}^s), \mathbf{v}^*), \sigma^2) + \log p_\theta(\mathbf{z}^s \mid \mathbf{z}^g) + \log p(\mathbf{z}^g) \propto \log p(\mathbf{z}^g, \mathbf{z}^s \mid \mathbf{x}^*, \mathbf{v}^*) \quad (7)$$

Each MH step first picks an object slot $i$ uniformly at random, then samples a new latents for that object from a proposal distribution $\tilde{p}(\mathbf{z}_i^{shape}, \mathbf{z}_i^{col})$, accepting/rejecting it according to the usual MH criterion [44]. The proposal distribution $\tilde{p}$ approximates $\frac{1}{J} \sum_{i=1}^{J} p(\mathbf{z}_i^{shape}, \mathbf{z}_i^{col})$ using a Gaussian mixture model fitted by expectation-maximisation [23]; it thus captures the distribution of object latent codes while disregarding the ordering of object indices. Note that the compositionality of our model increases efficiency of the chain, as it allows MH proposals that affect only one object while keeping other variables fixed – in contrast to a monolithic latent embedding, which would require accepting or rejecting global modifications to the entire scene. Thus, each MH step need not revert progress made on other variables: e.g. if the background is perfectly inferred but objects are not, then an MH proposal may change only an object, leaving the background intact. Also, it allows caching computation and only re-rendering parts of the scene that need to be considered for a proposed change (e.g. just background). In contrast, MCMC on non-structured models must render the entire scene from scratch.

## 4 Experiments

We conduct experiments on two synthetic datasets, using our model and three baselines. We also include ablations of our model, without MCMC inference, and with an unstructured latent space. We first evaluate performance in the standard setting where the test-set distribution matches the training distribution. Then, we evaluate generalisation to OOD data, by using several OOD test splits for each dataset, which have zero probability under the training distribution. We first describe these datasets and test splits (Sec. 4.1), the tasks, metrics and baselines used for evaluation (Sec. 4.2), and finally the performance of our proposed model, the baselines and ablations (Sec. 4.3). Implementation details for all models (hyperparameters, hardware, etc.) are given in the supplementary material.

### 4.1 Datasets

**GQN.** We render images of rooms containing several objects (cubes, cylinders, spheres), based on the 'rooms ring camera' dataset of [35]; similar datasets were used in [45, 33], but in all cases without OOD test splits. In the training split, the camera viewpoints are on a circular path around the center of the room, with the camera pointed at the center at fixed elevation angle. Textures for the walls and colors for the objects are selected randomly from a finite set, with some combinations held out. Three walls have the same texture as each other, with the fourth different. There are 3–4 objects present; these are placed near the side of the room identified by the odd texture. We define the following OOD test splits: (i) *position*: the 3–4 objects are now placed near a different wall; (ii) *composition*: unseen combinations of background and object textures; (iii) *number of objects*: 1/5/6 objects instead of

---

[2]Detection of the distribution shift could be done automatically with the generative model [13, 108]

[3]We also conducted early experiments with black-box variational inference [107], but this performed poorly due to becoming trapped in local minima. MCMC is more able to explore diverse modes of the posterior

| | GQN | | | | | | ARROW | | | | | |
|---|---|---|---|---|---|---|---|---|---|---|---|---|
| | per-image | | | | per-scene | | per-image | | | | per-scene | |
| | PSNR ↑ | D.MRE ↓ | ARI ↑ | mSC ↑ | PSNR ↑ | D.MRE ↓ | PSNR ↑ | D.MRE ↓ | ARI ↑ | mSC ↑ | PSNR ↑ | D.MRE ↓ |
| **Test** | | | | | | | | | | | | |
| Ours | 24.1 | 0.031 | 0.81 | 0.88 | 20.8 | 0.058 | 27.1 | **0.100** | **0.71** | **0.82** | 26.8 | **0.107** |
| Ours (w/o MCMC) [†] | 24.5 | 0.031 | 0.82 | **0.91** | 24.7 | 0.031 | 27.0 | 0.101 | 0.70 | **0.82** | 27.0 | 0.102 |
| Ours (w/o structure) | 32.1 | **0.016** | - | - | **27.8** | **0.030** | 22.2 | 0.990 | - | - | 20.1 | 0.990 |
| $\beta$-VAE | 20.6 | – | – | – | – | – | 24.3 | – | – | – | – | – |
| IODINE | 26.2 | – | 0.50 | 0.54 | – | – | 27.7 | – | 0.63 | 0.55 | – | – |
| Slot Att. | 30.5 | – | **0.94** | 0.67 | – | – | 28.3 | – | 0.48 | 0.17 | – | – |
| Slot Att. (MCMC) | 27.8 | – | 0.83 | 0.56 | – | – | 29.6 | – | 0.35 | 0.14 | – | – |
| uORF* | 27.0 | 0.027 | 0.74 | 0.59 | 24.2 | 0.049 | **35.1** | 0.176 | 0.64 | 0.44 | **33.8** | 0.202 |
| NeRF-VAE* | **32.0** | **0.016** | – | – | **27.8** | 0.033 | 25.3 | 0.991 | – | – | 25.3 | 0.991 |
| **OOD** | | | | | | | | | | | | |
| Ours | 21.8 | **0.034** | **0.68** | **0.89** | **18.3** | **0.069** | 26.7 | 0.139 | **0.57** | **0.81** | 26.2 | 0.137 |
| Ours (w/o MCMC) [†] | 11.4 | 0.170 | 0.44 | 0.55 | 11.5 | 0.176 | 21.0 | 0.160 | 0.40 | 0.61 | 21.0 | 0.159 |
| Ours (w/o structure) | 17.1 | 0.221 | - | - | 15.4 | 0.280 | 20.6 | 0.992 | - | - | 19.2 | 0.990 |
| $\beta$-VAE | 15.6 | – | – | – | – | – | 21.1 | – | – | – | – | – |
| IODINE | 19.7 | – | 0.44 | 0.53 | – | – | 25.0 | – | 0.42 | 0.42 | – | – |
| Slot Att. | 20.3 | – | 0.66 | 0.56 | – | – | 22.8 | – | 0.26 | 0.14 | – | – |
| Slot Att. (MCMC) | **23.7** | – | **0.68** | 0.54 | – | – | **27.2** | – | 0.32 | 0.17 | – | – |
| uORF* | 14.7 | 0.287 | 0.45 | 0.45 | 14.1 | 0.308 | 22.7 | **0.132** | 0.38 | 0.41 | 22.6 | **0.131** |
| NeRF-VAE* | 15.9 | 0.271 | – | – | 14.9 | 0.301 | 19.4 | 0.992 | – | – | 19.4 | 0.992 |

Table 1: Quantitative results on discriminative tasks, comparing performance for different methods on an in-distribution test set and OOD data. Dashes indicate the method does not support the task. Best results are shown in **bold**.

3–4; (iv) *viewpoint*: camera positions and elevations are randomly sampled. Exact details of the data generation process are given in the supplementary material.

**ARROW.** We render images using a modified version of the CLEVR dataset [59], similar to those in [57]. These have four objects, of which one is always an arrow, two of which are the same as each other, and a fourth that is different. The arrow always points at the odd (fourth) object. Object colors are randomly sampled. The camera has a random azimuth and shallow elevation. We define the following OOD test splits: (i) *position*: the four objects are positioned in a line, and the arrow no longer points to the odd object; (ii) *composition*: all objects are the same shape and color, with no arrow present; (iii) *number of objects*: 1/5/6 objects instead of four; the arrow still points at one odd object; (iv) *viewpoint*: camera looks down steeply on the objects.

## 4.2 Evaluation

**Tasks & metrics.** We consider the following discriminative tasks, all taking a single image as input: **instance segmentation**, measured by adjusted Rand index (**ARI**) and mean segmentation covering (**mSC**) [33], which in the PER-SCENE setting measures how well the model infers segmentation maps for novel viewpoints; **depth prediction**, measured by the mean relative error between predicted and true depths (**D.MRE**); and **pixel reconstruction**, measured by peak signal-noise ratio (**PSNR**), which in the PER-IMAGE setting measures how faithfully our latent representation can autoencode OOD images, and in the PER-SCENE setting how well the model performs novel view synthesis (NVS) We report these metrics according to two different protocols: (i) PER-IMAGE, where we calculate the metrics only on the input image passed to the model; and (ii) PER-SCENE, where we calculate the metrics jointly over 10 images of the scene, taken from different viewpoints, but still having received only one of these as input. Thus, the latter setting measures how well the model predicts appearance, depth and segmentation from novel viewpoints. Finally, we evaluate *a priori* image generation, measured by the Fréchet Inception distance (**FID**) [49] and kernel Inception distance (**KID**) [15] between sampled and ground-truth image distributions.

**Baselines.** We compare our approach to five existing works. **Slot Attention**[82] is a recent unsupervised object segmentation model, with a spatial mixture representation. It is purely 2D and not generative. We additionally experiment with MCMC inference on the trained Slot-Attention decoder (instead of using their encoder), to see whether the decoder generalises to OOD data. **uORF**[139] decomposes 2D images into 3D components represented as NeRFs. Unlike ours, it is not generative, and does not explicitly represent position separate from appearance. **NeRF-VAE**[72]

---

† The setting for this experiment is different, as at test-time the encoder requires the same number of images as used during training, whereas every other experiment inputs just one image

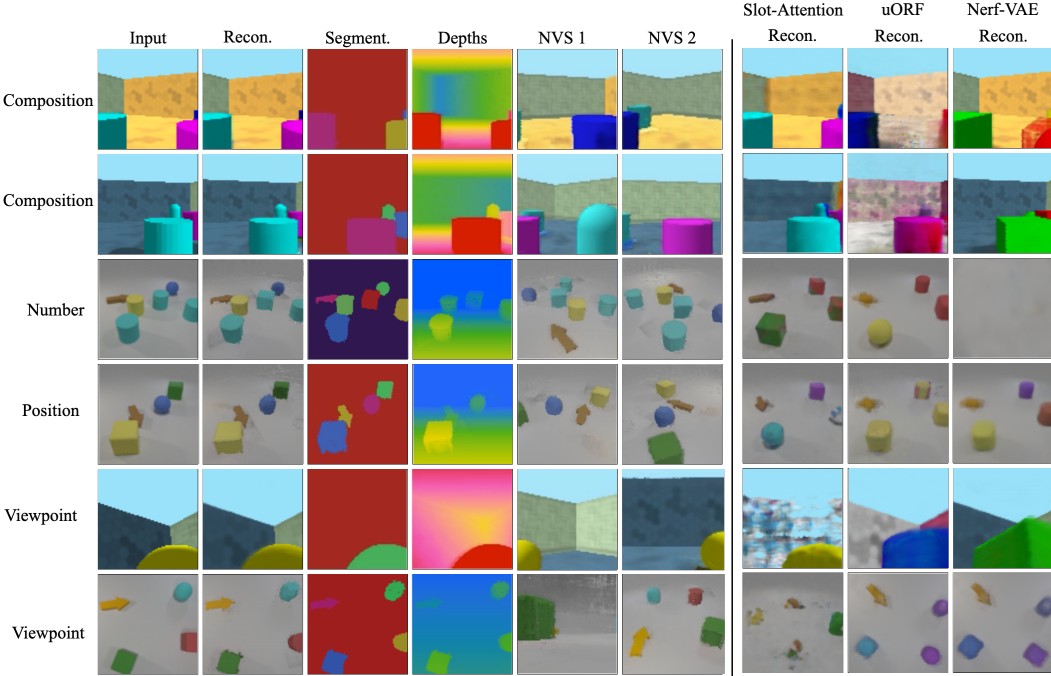

Figure 2: Qualitative results on various tasks, using our model and the baselines, on OOD data. Each row shows the input image, then (col. 2–6) outputs from our model: the reconstruction, instance segmentation, depth map, and two novel viewpoints. We see that our model predicts high-quality segmentations and depth-maps, and that the new viewpoints are plausible and consistent with the input. The final three columns show reconstructions (the easiest task) by the baselines. Here we see that they fail to generalise to OOD data – they have specialised to their training dataset, and learnt shortcuts [36] that interpret the input images as if they were drawn from it. Thus, they fail to map objects in unfamiliar viewpoints or contexts to appropriate latent representations.

is a generative method over 3D NeRFs, but which does not separate individual objects in its latent space. **IODINE**[40] is a discriminative 2D method which performs iterative amortized inference over a spatial mixture model. $\beta$-**VAE**[50] is an unstructured VAE aiming to learn disentangled representations, which has been hypothesized to help with OOD generalization. Note that the baselines are either discriminative approaches that do not support accounting for distribution shifts, or model the scene with one latent variable which cannot be intervened on to model a different distribution of scenes. Full details are in the supplementary Sec. 10.

## 4.3 Results

**In-distribution data.** We first evaluate how each model performs on the distribution of images it was trained on. Results on the discriminative tasks are given in Tab. 1; the top four rows show performance on the test split, which is drawn from the same distribution as the training data (all standard deviations are in the supplementary). Qualitative results are displayed in Fig. 2. We see that all methods successfully reconstruct input images (high per-image PSNR). The 3D-aware methods predict depths with a small relative error (low per-image D.MRE), with NeRF-VAE* slightly better on GQN, and ours on ARROW. Our method performs similarly to uORF*, with ours slightly better on segmentation but slightly worse for novel view synthesis. For GQN, Slot Attention performs best on segmentation under ARI scores, with ours better according to mSC and for both metrics on ARROW. Results for image generation are given in Tab. 2 and Fig. 3 (see the supplementary for images from NeRF-VAE*). Note that Slot Attention, IODINE and uORF* are excluded, as they cannot generate images *a priori*. Our model has clearly learnt the distribution of both datasets, including the space of likely object and background appearances. Quantitatively, ours performs similarly to NeRF-VAE* on GQN (with FID slightly better and KID slightly worse), and out-performs it on both metrics for ARROW. In Fig. 4, we also show that our semantically-interpretable latent space allows users to edit the scene shown in an image, by counterfactual inference over the model.

|  | GQN | | ARROW | |
| --- | --- | --- | --- | --- |
|  | FID | KID | FID | KID |
| Ours | **80.3** | 0.053 | **141.4** | **0.167** |
| Ours w/o $p_\theta(\mathbf{z}^s \mid \mathbf{z}^g)$ | 200.4 | 0.180 | 275.7 | 0.365 |
| NeRF-VAE* | 84.2 | **0.047** | 182.7 | 0.190 |

Table 2: Quantitative results on image generation for our method, ablation, and baseline.

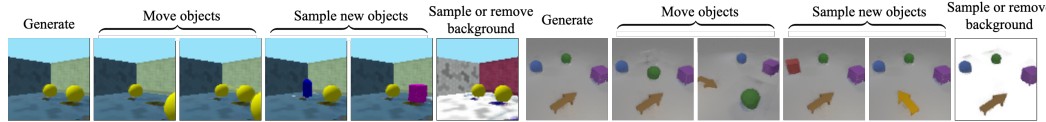

Figure 3: Examples of images sampled from our model for GQN (top) and ARROW (bottom)

Figure 4: Editing scenes using our model, by counterfactual inference. We condition on the input scene, then modify (intervene on) certain latent variables and re-render the result

**OOD data.** We now consider the more challenging setting where test data is drawn from a different distribution than the training set. Quantitative results in this setting are given in the bottom half of Tab. 1. This shows the mean over all OOD splits; a full breakdown is given in the supplementary. We see that in general, our method significantly out-performs the baselines on these splits – showing the benefit of our causal model and inference scheme. In particular, our method performs best on all combinations of vision tasks and datasets, except for depth estimation on ARROW where uORF* is slightly better. Slot-Attention with MCMC and IODINE with iterative amortized inference successfully reconstruct input images during their test-time optimization, but this comes at the cost of lower segmentation performance. In contrast, our method performs well on both reconstruction and segmentation. Thus, the best methods on the in-distribution test split are *not* the best on OOD data. They leverage features that are predictive of the output in the training set [36] and easy to learn [133, 117, 106], but which are not reliable across different distributions [91]. Our qualitative results (Fig. 2) reinforce this interpretation – we see that our method still predicts accurate segmentations and depth-maps for OOD data, and novel viewpoints look plausible. Notably, in all cases, it successfully reconstructs the input image via its latent space and synthesises new viewpoints, in spite of never having seen such an image during training. In contrast, the baselines struggle with images that are different to their training distribution – e.g. mispredicting object colors (1st row, uORF* and NeRF-VAE*), predicting an over-smoothed average scene (3rd row, NeRF-VAE*), or failing to separate walls and ceiling when they are in an unfamiliar pose (5th row, Slot Attention).

In the supplementary material (Sec. 6), we provide a more detailed analysis of how model performance generalizes on various axes of out-of-distribution images. Particularly noteworthy is that the 2D baseline Slot Attention [82] has the most significant drop in reconstruction performance (measured by PSNR) on images taken from out-of-distribution camera viewpoints (Tab. 3) In contrast, our 3D-aware model has a much smaller drop in performance. Similarly important are the qualitative results in Fig. 5, showing that the non-compositional baseline model NeRF-VAE [72] fails on the straightforward task of reconstructing the input image – it predicts scenes similar to ones seen in the training dataset even when the OOD images are clearly different (e.g. still predicting the same object positions as in the training set despite objects being at the opposite side of the room). In contrast, our method which uses test-time optimization accurately reconstructs out-of-distribution input images.

## 4.4 Ablation Study

**Ablating MCMC inference.** We perform an ablation study on the effects of our novel MCMC scheme. We compare it with the common approach of amortised inference (i.e. an encoder network predicts the posterior parameters). This is denoted in Tab. 1 as *Ours (w/o MCMC)*. We see that amortised inference is indeed a critical bottleneck for out-of-distribution generalization: though it performs well when the test distribution is identical to the training distribution, its performance drops significantly on out-of-distribution images, while MCMC holds up.

**Ablating generative model structure.** We analyse the effects of using our compositional model compared to a non-compositional model, which has one global latent variable rather than one per object (similar to NeRF-VAE), but otherwise with the same architecture and inference as ours. This

is denoted in Tab. 1 as *Ours (w/o structure)*. We see that our proposed compositional generative model significantly improves out-of-distribution generalization over the unstructured ablation: though both perform similarly on IID test data, the unstructured model performs significantly worse on out-of-distribution images.

**Ablating** $p_\theta(\mathbf{z}^s | \mathbf{z}^g)$**.** We analyse the effects of our high-level prior over scene variables. In Tab. 2, we evaluate samples generated by the model, comparing FID and KID with an ablated model that samples $\mathbf{z}^s$ from a fixed prior. The poorer scores from the ablated model demonstrate that our hierarchical approach with a high-level prior is necessary to correctly model the density of scenes: our model samples plausible scenes as it can model relationships between objects, while the ablated model performs much worse.

### 4.5 Limitations and Societal Impacts

Currently our method has several limitations.

- The scene-level prior distribution $p_\theta(\mathbf{z}^s \mid \mathbf{z}^g)$ uses an unstructured latent space and decoder – thus, this particular component of the model is not interpretable. It would be worthwhile to use a structured prior, that explicitly models sparse relations among objects.

- We have only tested it on synthetic data; it would be interesting to conduct experiments on natural images. This would require modelling the image-formation process more faithfully, e.g. by modelling light transport and non-Lambertian materials.

- Our MCMC inference scheme is computationally expensive, as we render a complete image at each step. The efficiency could be increased by using a stochastic gradient for the LD transitions.

- Our model lacks variables to explicitly model object presence – if an object slot is unused, it learns a latent representation of an empty, zero-density object.

- It would be preferable to use less dataset-specific prior information, e.g. candidate cells for objects.

- Optimizing the model required multiple stages of training, and optimising Eq. 5 was unstable. It would be worthwhile to investigate joint training as in [57].

We do not anticipate any negative societal impacts from this work. While all generative models have the potential to be used for creating fake content, our method requires further development to work on realistic images. On discriminative tasks, support for OOD inference should reduce the susceptibility of models to dataset bias; thus, we see a significant long-term benefit to such methods.

## 5 Conclusion

We have presented a new object-centric generative model that captures the causal process by which images are produced, and incorporates universal physical knowledge such as 3D geometry. We have shown that it is possible to intervene on our model then perform Monte-Carlo inference, in order to process OOD images. On instance segmentation and novel view synthesis in this OOD setting, it significantly out-performs three state-of-the-art approaches.

## Acknowledgments and Disclosure of Funding

We thank Professor Christopher K. I. Williams for detailed feedback on the draft and Professor Iasonas Kokkinos and Hakan Bilen for fruitful discussions. We also thank the NeurIPS reviewers for extensive feedback, discussion and experiment suggestions. This work was mostly done during T.A.'s internship at Snap Inc. T.A. was partly supported by an EPSRC Doctoral Training Partnership.

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
