## Supplementary Material

## 6 Additional Results

Here we report results for each out-of-distribution (OOD) dataset split separately. In Tab. 3, we report all metrics as described in the main paper (Sec. 4.2), also including standard deviations. We also show quantitative results in Fig. 5 to compare and contrast the performance of all models on GQN dataset.

Tab. 3 shows that our method significantly outperforms Slot-Attention, uORF* and NeRF-VAE* on most metrics. In cases of a worse performance, these are mostly within one standard deviation, hence not statistically significant. Particularly noteworthy is the significant improvement in the challenging OOD viewpoint setting – a dataset containing images taken from a radically different viewpoint than those seen during training. As reported in prior works [4, 51, 8, 71, 7], such OOD images produce very different feature representations in a standard neural network that has no generalization guarantees; this does not apply to our 'generative causal understanding' framework since it does not use amortised inference (e.g. encoder network) but instead performs inference directly over the causal generative model. We see that our model achieves higher PSNR and segmentation scores than the other baselines. It is noteworthy that the 2D baseline Slot-Attention [82] has a very significant drop in performance across all metrics for this particular data split, demonstrating that it does not learn to generalize to OOD camera viewpoints on our datasets. Qualitative results in Fig. 5 (rows 7 and 8) similarly show that Slot-Attention [82] fails to reconstruct the input image, performing significantly worse than 3D-aware methods (ours, NeRF-VAE* and uORF*).

Our model also achieves the best reconstruction (PSNR) and segmentation (mSC) performance on the OOD composition dataset. It is noteworthy that 2D compositional discriminative method (Slot-Attention [82]) performs well on segmentation task (best ARI) and is among the best on generalization to OOD number of objects (consistent with the Slot-Attention paper), though it cannot perform 3D reasoning tasks. On the other hand, less compositional (NeRF-VAE*) and discriminative (uORF*) baselines show a significant drop in performance on images containing OOD compositions – we hypothesize this is because they do not learn to represent such 3D scenes. This is based on a quantitative results in Fig. 5, which illustrates that baselines (last two columns) fail on even the relatively straightforward task of reconstructing the input image, when that image is out-of-distribution. Note that uORF* continues to predict scenes that are similar to those in the training set, despite the inputs being significantly different. For example, in the first two rows, uORF* incorrectly outputs objects with a colour that were present with an orange background in the training set. This is in line with other reports in the literature (e.g. [36]) showing that potentially the baseline has learnt a 'shortcut', to presumptively output a red object when the background pixels are orange. Similarly, uORF* predicts incorrect positions of objects in the OOD position split (rows 5 and 6) – in fact, predicting them on the opposite side of the room (as they appear in the training set). In contrast, our model successfully localises objects – we attribute this to our inference mechanism and causal model which represents object with appearance disentangled from position.

In Fig. 6–8, we show generation results for our proposed method and for NeRF-VAE*. These are examples of scenes sampled *a priori*, rendered from multiple viewpoints. We see that in accordance with the quantitative results in the main text, our model is able to sample significantly more realistic scenes (and thus images) than NeRF-VAE*.

| | GQN | | | | | | ARROW | | | | | |
|---|---|---|---|---|---|---|---|---|---|---|---|---|
| | per-image | | | | per-scene | | per-image | | | | per-scene | |
| | PSNR ↑ | D.MRE ↓ | ARI ↑ | mSC ↑ | PSNR ↑ | D.MRE ↓ | PSNR ↑ | D.MRE ↓ | ARI ↑ | mSC ↑ | PSNR ↑ | D.MRE ↓ |
| **composition** | | | | | | | | | | | | |
| Ours | **23.3 ± 2.9** | **0.03 ± 0.003** | 0.82 ± 0.22 | **0.9 ± 0.07** | **19.6 ± 1.9** | **0.067 ± 0.047** | **26.5 ± 0.9** | 0.09 ± 0.018 | **0.73 ± 0.06** | **0.82 ± 0.02** | **26.1 ± 0.9** | 0.09 ± 0.007 |
| Slot Att. | 23.1 ± 2.2 | – | **0.88 ± 0.2** | 0.63 ± 0.11 | – | – | 25 ± 2.3 | – | 0.27 ± 0.14 | 0.13 ± 0.03 | – | – |
| uORF* | 13.6 ± 3 | 0.18 ± 0.158 | 0.43 ± 0.3 | 0.43 ± 0.15 | 13.3 ± 2 | 0.208 ± 0.105 | 25.3 ± 1.7 | **0.083 ± 0.019** | 0.57 ± 0.14 | 0.39 ± 0.12 | 25.4 ± 1.4 | **0.083 ± 0.006** |
| NeRF-VAE* | 15.1 ± 3.2 | 0.074 ± 0.101 | – | – | 14.5 ± 1.5 | 0.114 ± 0.069 | 19.2 ± 1.6 | 0.993 ± 0 | – | – | 19.2 ± 1.2 | 0.993 ± 0.0003 |
| **position** | | | | | | | | | | | | |
| Ours | 21.7 ± 2.5 | **0.033 ± 0.008** | **0.72 ± 0.3** | **0.86 ± 0.09** | 19.2 ± 2.2 | 0.083 ± 0.079 | **26.5 ± 0.6** | 0.086 ± 0.015 | **0.69 ± 0.05** | **0.8 ± 0.03** | **26.3 ± 0.5** | 0.085 ± 0.006 |
| Slot Att. | **23.4 ± 3.1** | – | 0.67 ± 0.34 | 0.63 ± 0.14 | – | – | 21.4 ± 0.9 | – | 0.36 ± 0.17 | 0.14 ± 0.03 | – | – |
| uORF* | 12.7 ± 2.7 | 0.281 ± 0.226 | 0.14 ± 0.18 | 0.26 ± 0.12 | 12.5 ± 2.2 | 0.34 ± 0.132 | 20.3 ± 1 | **0.076 ± 0.016** | 0.32 ± 0.17 | 0.28 ± 0.05 | 20.4 ± 0.8 | **0.076 ± 0.003** |
| NeRF-VAE* | 12.7 ± 2.7 | 0.267 ± 0.225 | – | – | 12.6 ± 2.2 | 0.329 ± 0.136 | 19.5 ± 1.2 | 0.993 ± 0 | – | – | 19.5 ± 0.8 | 0.993 ± 0.0003 |
| **number** | | | | | | | | | | | | |
| Ours | 23.1 ± 3 | **0.034 ± 0.01** | 0.67 ± 0.19 | **0.87 ± 0.08** | **19.4 ± 2.4** | **0.063 ± 0.031** | **27.1 ± 0.8** | 0.087 ± 0.017 | **0.46 ± 0.04** | **0.83 ± 0.02** | **26.8 ± 0.5** | 0.087 ± 0.006 |
| Slot Att. | **24.0 ± 2.4** | – | **0.74 ± 0.27** | 0.6 ± 0.11 | – | – | 23.8 ± 1.5 | – | 0.27 ± 0.13 | 0.16 ± 0.03 | – | – |
| uORF* | 18.4 ± 3.2 | 0.128 ± 0.097 | 0.49 ± 0.33 | 0.44 ± 0.15 | 17.3 ± 2.1 | 0.14 ± 0.045 | 22.7 ± 1.3 | **0.081 ± 0.018** | 0.29 ± 0.21 | 0.45 ± 0.09 | 22.7 ± 1 | **0.081 ± 0.004** |
| NeRF-VAE* | 20.5 ± 3.9 | 0.094 ± 0.067 | – | – | 18.7 ± 2.5 | 0.114 ± 0.032 | 19.5 ± 1.5 | 0.993 ± 0 | – | – | 19.6 ± 0.9 | 0.993 ± 0.0003 |
| **viewpoint** | | | | | | | | | | | | |
| Ours | **19.0 ± 4.4** | **0.035 ± 0.051** | 0.61 ± 0.44 | **0.91 ± 0.1** | **15.5 ± 2.5** | **0.072 ± 0.059** | 25.6 ± 3.1 | 0.395 ± 0.701 | **0.64 ± 0.12** | **0.77 ± 0.08** | 24.4 ± 1 | 0.389 ± 0.007 |
| Slot Att. | 11.7 ± 1.7 | – | 0.42 ± 0.44 | 0.45 ± 0.15 | – | – | 19 ± 1.8 | – | 0.16 ± 0.09 | 0.11 ± 0.02 | – | – |
| uORF* | 10.7 ± 3.6 | 0.581 ± 0.301 | 0.57 ± 0.48 | 0.58 ± 0.27 | 10.5 ± 2.6 | 0.593 ± 0.195 | 22.5 ± 3.3 | **0.39 ± 0.679** | 0.52 ± 0.19 | 0.4 ± 0.1 | 22.2 ± 1.1 | **0.382 ± 0.009** |
| NeRF-VAE* | 10.8 ± 3.4 | 0.636 ± 0.246 | – | – | 10.5 ± 2.1 | 0.662 ± 0.058 | 19.3 ± 1.6 | 0.985 ± 0.008 | – | – | 18.9 ± 0.6 | 0.985 ± 0.0001 |

Table 3: Quantitative results on discriminative tasks, comparing performance for different methods on OOD data splits. Dashes indicate the method does not support the task.

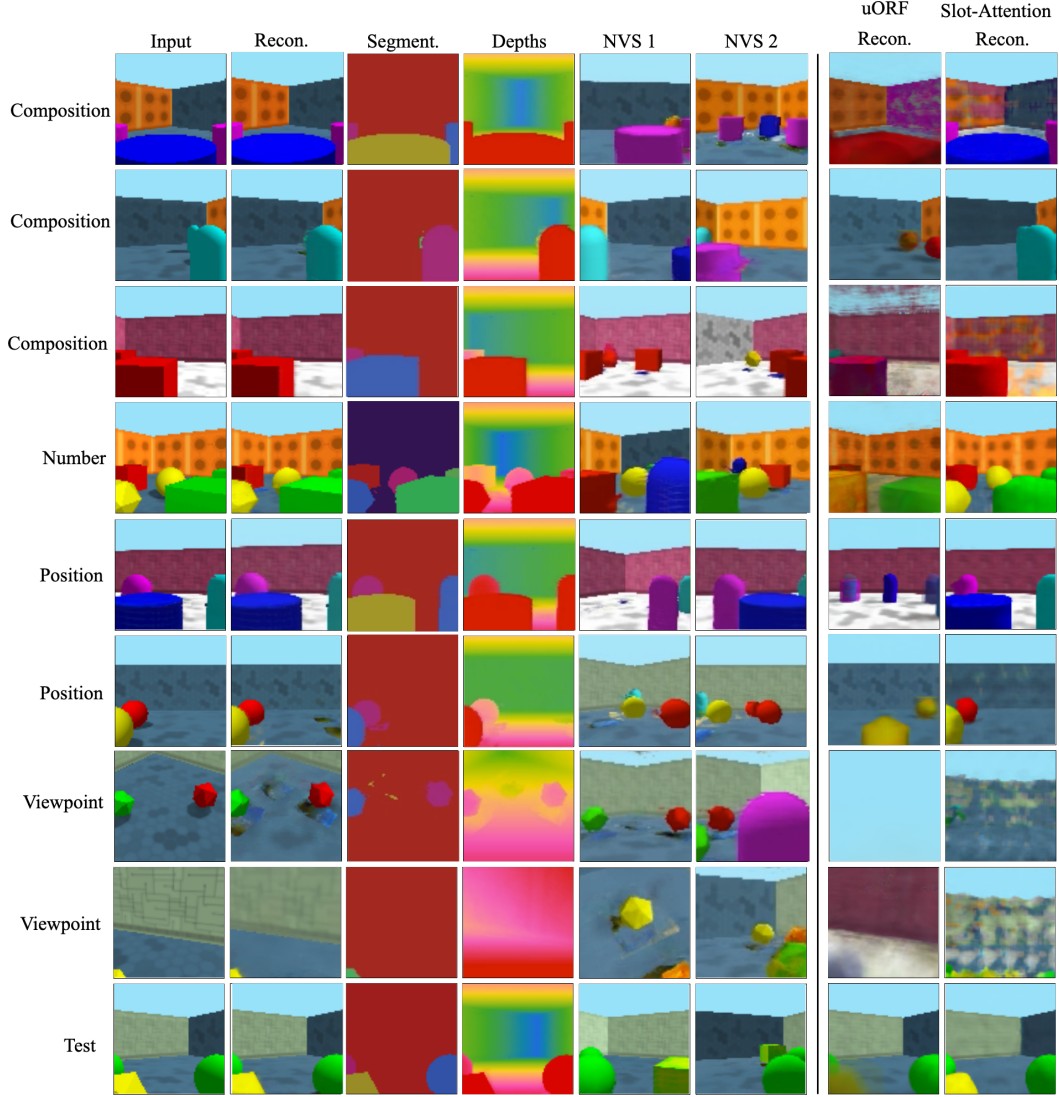

Figure 5: Additional qualitative results from our method and baselines uORF* and Slot Attention. See the main paper for a discussion of tasks and metrics, and supplementary Sec. 6 for discussion of successes and failures

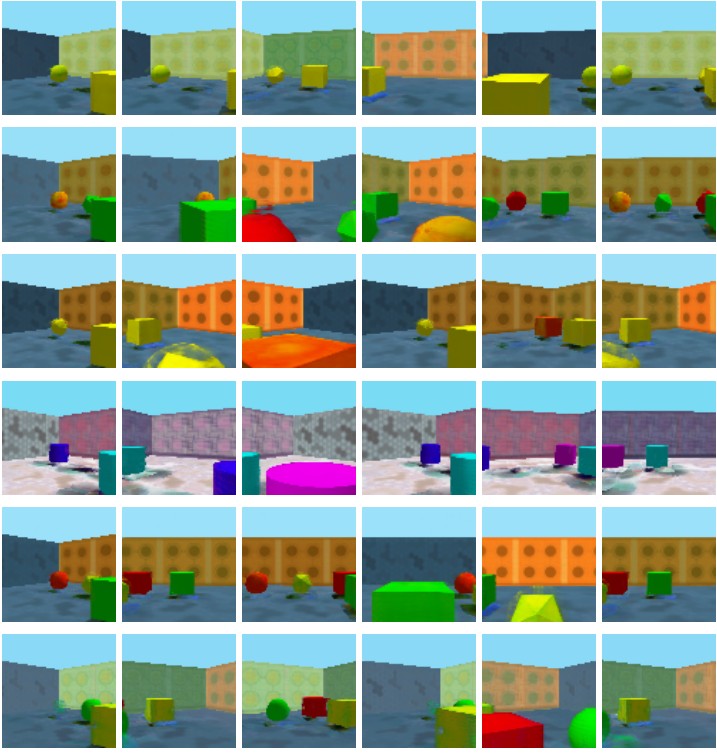

Figure 6: Additional generation results from our method on GQN. Each row corresponds to a single scene, with each column showing a different viewpoint

# 7 Architecture Details

We illustrate the probabilistic graphical model for our approach in Fig. 9. In the remainder of this section, we specify the architectures of the neural networks that parameterize the various conditional distributions.

The variational posterior $q_\phi(\mathbf{z}^s \mid \{\mathbf{x}_n, \mathbf{v}_n\}_{n=1}^M)$ used during training is parameterised by an image encoder, pose encoder and pooled representation encoder. For the image encoder, we use a convolutional neural network:

| Layer | Filters | Stride | Norm./Act. |
|---|---|---|---|
| Conv $4 \times 4$ | 16 | 1 | Layer/CELU |
| Conv $3 \times 3$ | 16 | 1 | Layer/CELU |
| Conv $4 \times 4$ | 32 | 2 | Layer/CELU |
| Conv $3 \times 3$ | 32 | 1 | Layer/CELU |
| Conv $4 \times 4$ | 64 | 2 | Layer/CELU |
| Conv $3 \times 3$ | 64 | 1 | Layer/CELU |
| Conv $4 \times 4$ | 128 | 2 | Layer/CELU |
| Conv $3 \times 3$ | 128 | 1 | Layer/CELU |
| Conv $4 \times 4$ | 128 | 2 | Layer/CELU |

For the camera pose encoder, we use a fully-connected neural network:

| Layer | Size | Norm./Act. |
|---|---|---|
| MLP | 50 | Layer/CELU |
| MLP | 50 | Layer/CELU |
| MLP | 50 | Layer/CELU |
| MLP | 50 | Layer/CELU |
| MLP | 50 | Layer/CELU |
| MLP | 50 | – |

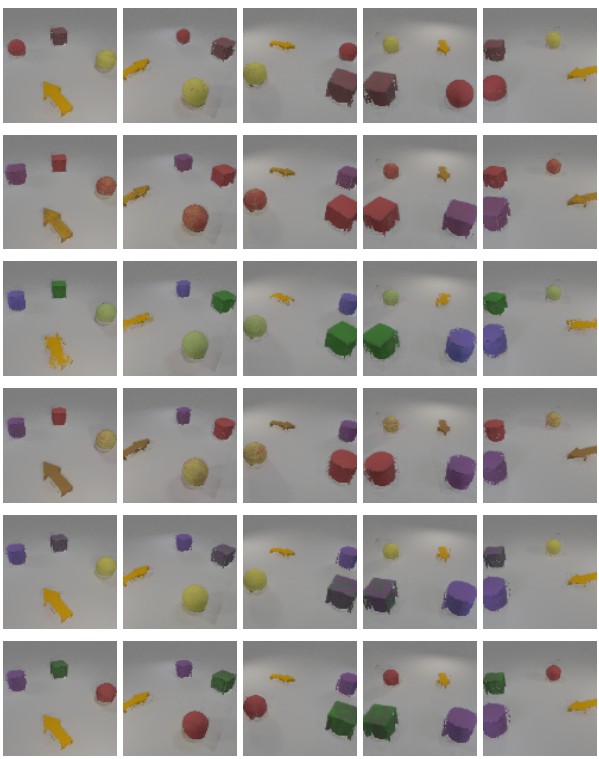

Figure 7: Additional generation results from our method on ARROW. Each row corresponds to a single scene, with each column showing a different viewpoint

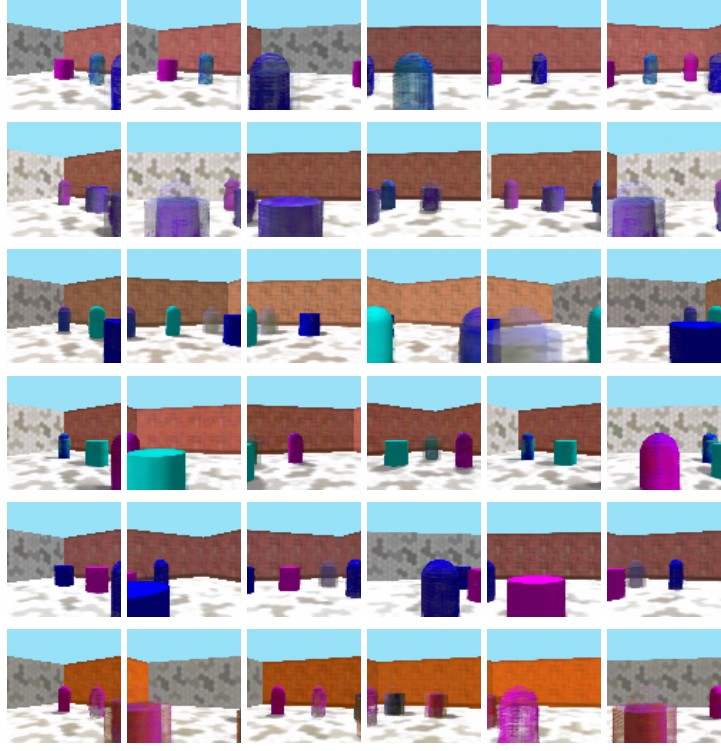

Figure 8: Generation results from NeRF-VAE* on GQN. Each row corresponds to a single scene, with each column showing a different viewpoint

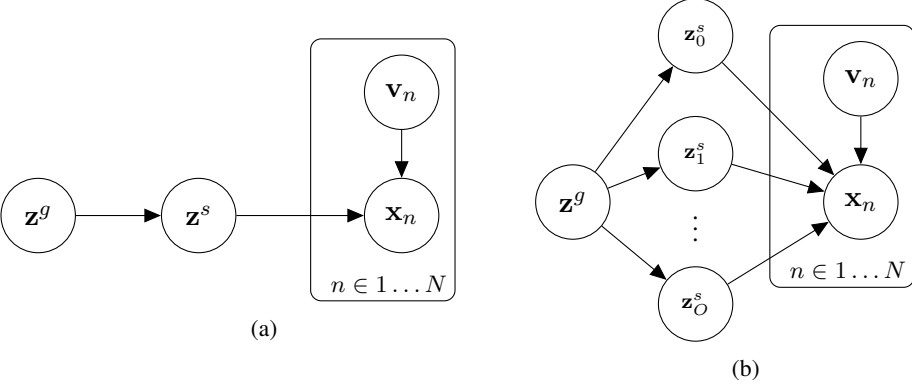

(a)

(b)

Figure 9: The probabilistic graphical model used in our approach, in simplified and detailed form. In (a), we group all the object-level latents (representing shape, color and position) into a single node for clarity. In (b) we show the full factorization present in the model; note that each object has different distributions for the latents, depending on $\mathbf{z}^g$ via the networks $\zeta_\theta$ and $\xi_\theta$. For (b) $\mathbf{z}_0^s$ corresponds to background $\{\mathbf{z}_{bg}^{shape}, \mathbf{z}_{bg}^{col}\}$ and $\mathbf{z}_i^s$ to $i$-th object $\{\mathbf{z}_i^{shape}, \mathbf{z}_i^{col}, \mathbf{z}_i^{pos}\}$

We then pool the image and camera representations – we sum-pool representations of $M = 5$ images with their corresponding viewpoints $\{\mathbf{x}_n, \mathbf{v}_n\}_{n=1}^M$ for GQN dataset and use $M = 1$ for Arrow dataset. The pooled representation is taken as input to the scene representation encoder, which outputs shape and appearance embeddings for each component, and an embedding $k$ for each object's position:

| Layer | Size | Norm./Act. |
|-------|------|------------|
| MLP | 116 | Layer/CELU |
| MLP | 116 | Layer/CELU |
| MLP | 116 | Layer/CELU |

The object position embedding $k$ is then passed through another fully-connected neural network to output logits for a Gumbel Softmax (relaxed categorical variable) one-hot position indicator.

For GQN and Arrow datasets, object shape and appearance embeddings are both 9 dimensional. Background appearance is 3 dimensional for GQN whilst we use 1 dimension for Arrow. In a second stage, we variationally autoencode object-level scene representation to a global scene variable $\mathbf{z}^g$. $\mathbf{z}^g$ is a Gaussian variable with 10 dimensions for Arrow and 20 dimensions for GQN. Encoder and decoder are fully-connected residual neural networks with the following layers:

For Arrow:

| Layer | Size | Norm./Act. | Residual Connection |
|-------|------|------------|---------------------|
| MLP | 300 | Layer/CELU | No |
| MLP | 300 | Layer/CELU | Yes |
| MLP | 300 | Layer/CELU | Yes |
| MLP | 300 | Layer/CELU | Yes |
| MLP | 300 | Layer/CELU | Yes |
| MLP | 300 | Layer/CELU | Yes |
| MLP | 300 | Layer/CELU | Yes |

For GQN:

| Layer | Size | Norm./Act. | Residual Connection |
|-------|------|------------|---------------------|
| MLP | 400 | Layer/CELU | No |
| MLP | 400 | Layer/CELU | Yes |
| MLP | 400 | Layer/CELU | Yes |
| MLP | 400 | Layer/CELU | Yes |
| MLP | 400 | Layer/CELU | Yes |
| MLP | 400 | Layer/CELU | Yes |
| MLP | 400 | Layer/CELU | Yes |
| MLP | 400 | Layer/CELU | Yes |
| MLP | 400 | Layer/CELU | Yes |
| MLP | 400 | Layer/CELU | Yes |

The 3D shape and appearance of the object and background components are represented as Neural Radiance Fields (NeRFs) [88, 137, 97], conditioned on the component's appearance and shape embeddings. We define two NeRF MLPs – one for background component and one for object component. As shown in the following Fig. 10, to facilitate disentanglement of shape and appearance, the opacity $\sigma$ depends only on shape embedding, while the radiance $\mathbf{c}$ depends only on appearance embedding.

Aside from these conditioning vectors, we use the vanilla NeRF architecture used in [88]. During volumetric rendering, each 3D point is passed through a standard positional embedding which outputs

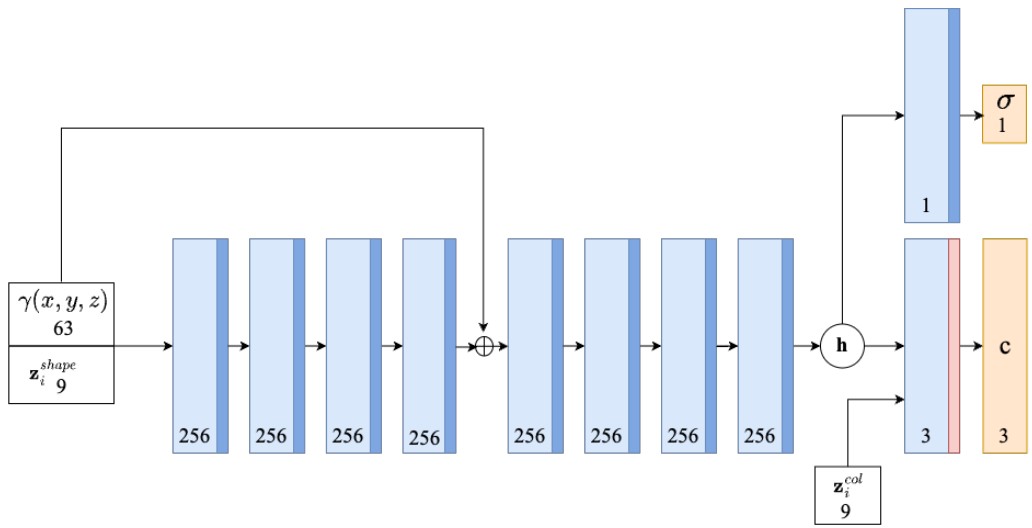

Figure 10: Architecture of a neural radiance field [88, 137, 97] conditioned on component's shape and appearance codes.

a 63-dimensional vector. Then the first part of the network takes a positional embedding of 3D point $\gamma(x, y, z)$ and a shape embedding and passes it through 8 fully-connected layers to output a feature vector $h$:

| Layer | Size | Act. |
|-------|------|------|
| MLP | 256 | ReLU |
| MLP | 256 | ReLU |
| MLP | 256 | ReLU |
| MLP | 256 | ReLU |
| MLP | 256 | ReLU |
| MLP | 256 | ReLU |
| MLP | 256 | ReLU |
| MLP | 256 | ReLU |

Then $h$ is passed through a linear layer with ReLU activation to output the density $\sigma$. Then $h$ concatenated with component's appearance embedding is passed through a linear layer with sigmoid activation to output the radiance $\mathbf{c}$. We initialise the network with a bias of $3.0$ before the output of density $\sigma$. During training we add Gaussian noise with standard deviation of $0.3$ to the output of $\sigma$.

## 8 Implementation Details

Our final models were trained on a single NVIDIA A100 GPU for approximately 4 weeks (we did not measure the total compute time for development). We implemented our model with PyTorch [101]. We train our model with Adam [64] using a learning rate of $1e-4$. During the first stage of training, we use a batch size of 8. During training, we subsample and render 8000 pixels, and for each pixel, we sample $512$ 3D points along a ray (we use 256 points initially to speed up training) from a distance of $0.01$ to $20.0$ for Arrow and from $0.01$ to $9.9$ for GQN. Other rendering hyperparameters are identical to those of [88, 137]. During a second stage of training, we use batch size of 64. We multiply the KL-divergence term in ELBO loss by $\beta$ [50]: we use $\beta = 50$ for GQN and $\beta = 10$ for Arrow.

During training, foreground is modelled with 5 object components and one component is used for modelling the background. During test-time, the number of object components is fixed to the maximum number of objects present in any dataset split. During both train and test time, we provide the model with possible object candidate positions [45, 20] and fix the background component to be outside the possible foreground range in a dataset [45, 139, 122]. We have initially performed experiments without such inductive biases, finding that model performs well only in around 1 in 10 runs; with the rest of runs model captures background with foreground components and vice-versa.

Rendering a scene during training assumed that a random light is being emitted from the world (in contrast to assumption of black or white world in [88]) – this prevents model from exploiting the world's emitted colour to model the scene and encourages modelling non-trivial scene to avoid random light from entering the camera.

The Gaussian mixture model is trained by Expectation Maximisation [23]. Each component has its own general covariance matrix. For GQN, we use 20 Gaussian mixture model components for background variables and 20 components for the object variables. For Arrow, we use 5 components for background variables and 70 components for the object variables.

The image is modelled with a Normal distribution with a mean outputted by a model as described in the method Section Sec. 3.1 and a standard deviation of $0.15$. During MCMC inference, the gradient descent step has a learning rate of $1e-4$. The variatiotional autoencoder models their output with a Gaussian distribution with a fixed standard deviation of $0.01$. In practice we perform ten LD steps followed by a single MH step, and repeat this to convergence or until 20 thousand iterations are reached. When evaluating results, we take the posterior mode, i.e. the single sample from each chain that had maximum probability.

We performed hyperparameter optimization with a grid search over the learning rate $(1e-5, 3e-5, 5e-5, 7e-5, 9e-5, 1e-4, 3e-4, 5e-4, 7e-4, 9e-4, 1e-3, 3e-3, 5e-3, 7e-3, 9e-3)$, batch size $(1, 2, 4, 8, 16, 32, 64, 128)$, number of subsampled/rendered pixels used for estimating the loss $(4000, 8000, 12000, 16000, 20000)$, number of sampled 3D points along a ray $(64, 128, 256, 512)$.

## 9 Datasets

We now describe the datasets that we used for training and evaluation.

### 9.1 GQN

We render images of shape $(80, 80, 3)$ of rooms containing several objects (icosahedrons, cubes, capsules, cylinders, spheres), based on the 'rooms ring camera' dataset of [35]; similar datasets were used in [45, 33], but in all cases without OOD test splits. For evaluation with each OOD dataset split, we sample one image from each scene and use the sampled image for per-image metrics while using all 30 for per-scene metrics. The data generation source code will be made public.

**Training split.**    The training split contains 10000 scenes, each with 30 RGB images. The camera viewpoints are on a circular path around the center of the room with the elevation of $1.0$ and with the camera pointed at the center of the room at the same elevation angle. The first camera viewpoint along circular path is generated by first sampling a random initial yaw of the camera with respect to the origin of the scene and then shifting a camera in $xy$ (horizontal) plane by a random distance (sampled uniformly from $\mathcal{U}_{[3.1, 3.4]}$) from the origin. Subsequent viewpoints form a circular path with respect to the origin. Textures for the walls and colors for the objects are selected randomly from a finite set, with some combinations held out. Three walls have the same texture as each other, with the fourth different. In particular, the training split contains scenes with either (i) random compositions of (light, cerise) background textures with (capsule, cylinder) objects with 3 random colours or (ii) random compositions of (cosahedron, box, sphere) objects with another 3 random colours and with (orange, blue, green) background textures. There are 3–4 objects present; these are placed near the side of the room identified by an odd background texture.

**OOD composition split.**    The OOD composition split contains 100 scenes, each with 30 RGB images. The data generation process is same as for training split, except that we swap possible background textures. Hence, the split contains scenes with either (i) random compositions of (orange, blue, green) background textures with (capsule, cylinder) objects with 3 random colours or (ii) random compositions of (cosahedron, box, sphere) objects with another 3 random colours with (light, cerise) background textures.

**OOD position split.**    The OOD position split contains 100 scenes, each with 30 RGB images. The data generation process is same as for training split, except that objects are now placed in the opposite side of the room identified with an odd background texture.

**OOD viewpoint split.**    The data generation process is same as for training split, except that camera viewpoint is sampled from a different procedure. In particular, we have two OOD viewpoint splits, each containing 100 scenes with 30 RGB images. The first part contains camera elevation sampled from uniform distribution $\mathcal{U}_{[0.1,4.0]}$ and position sampled randomly in between objects. The camera has a random yaw and its pitch is such that the camera is focused on the point with elevation of 1.0 at a distance of 1.0. A second part of OOD viewpoint split contains images with camera pointing from high-up to the center of the room with a circular path around the origin. For each frame, the camera has a $xy$ (horizontal) distance of 1.5 from the origin, elevation of 2.5 and a pitch of $-0.75$.

**OOD number split.**    The OOD number split contains 300 scenes, each with 30 RGB images. The data generation process is same as for training split, except that we now have 1, 5 or 6 objects (100 scenes per each) instead of 3-4 objects. Objects are placed randomly in the opposite side of the room identified with an odd background texture, but in contrast to training split, now some objects can be placed in the middle of the room to avoid cluttering the scene.

### 9.2    ARROW

We render RGB images of shape $(96, 96, 3)$ using a modified version of the CLEVR dataset [59], similar to those in [57]. Similarly to [72, 93], we modify background component such that the background is present from all camera viewpoints.

**Training split.**    The training dataset contains 10000 scenes, with 20 images per each scene. Each scene has four objects, of which one is always an arrow, two of which are the same as each other, and a fourth that is different. The arrow always points at the odd-shaped (fourth) object. The colours of objects are randomly sampled. The camera points to the origin $(0, 0, 0)$ from two circular path around it: one starting at position of $(4.99, -4.33, 4.89)$, a second starting at $(3.75, 3.75, 1.0)$.

**OOD composition split.**    The OOD composition dataset contains 100 scenes, with 10 images per each scene. For evaluation, we sample one image per scene and use it to evaluate per-image metric while using all 10 images for per-scene metrics. The data generation process is same as for training split, except that now all objects contain same shape and colour, with no arrow present.

**OOD position split.**    The OOD position dataset contains 100 scenes, with 10 images per each scene. For evaluation, we sample one image per scene and use it to evaluate per-image metric while using all 10 images for per-scene metrics. The data generation process is same as for training split, except that now four objects are positioned in a an approximate line, and the arrow no longer points to an odd object.

**OOD viewpoint split.**    The OOD position dataset contains 100 scenes, with 21 images per each scene. It contains one image rendered from the top at position $(0., 0., 8.0)$ to the origin $(0, 0, 0)$. The other 20 images are from two circular paths around the origin: one starting at position of $(4.75, 4.75, 0.4)$, a second starting at $(0.5, 0.5, 8.0)$. Both starting positions are with added noise from uniform distribution $\mathcal{U}_{[-0.5, 0.5]}$ to each dimension.

**OOD number split.**    The OOD position dataset contains 100 scenes, with 10 images per each scene. The data generation process is same as for training split, but now the scene contains 1, 5 or 6 objects instead of 4.

## 10    Baselines

For Slot-Attention [82], we used the original publicly-available implementation, with hyperparameters (including number of slots and number of iterations) re-tuned on our datasets. For the $80 \times 80$ GQN images, we slightly modified the decoder architecture, increasing the initial feature map size. As for our method, at test time, we set the number of slots equal to the largest number of objects (plus one for background) in any OOD split.

We reimplemented [139, 72] in our own framework, and denote these as uORF* and NeRF-VAE* respectively. For a fair comparison, we use the same number of samples along each ray, the same or larger number of parameters, and similar encoder/decoder architectures as for our own model. For a

| | AIR [34] | GQN [35] | MONET [17] | IODINE [40] | GENESIS [33] | SCALOR [58] | SPACE [78] | OCF [5] | GNM [57] | ROOTS [20] | RELATE [30] | MuLMON [93] | ECON [128] | O3V [45] | GENESIS-v2 [32] | GIRAFFE [97] | uORF [139] | NeRF-VAE [72] | SIMONE [60] | ObSuRF [122] | *ours* |
|---|---|---|---|---|---|---|---|---|---|---|---|---|---|---|---|---|---|---|---|---|---|
| *explicitly represents 3D shapes* | ✗ | ✗ | ✗ | ✗ | ✗ | ✗ | ✗ | ✗ | ✗ | ✗ | ✗ | ✗ | ✗ | ✓ | ✗ | ✓ | ✓ | ✓ | ✗ | ✓ | ✓ |
| *explicitly represents 3D object positions* | ✗ | ✗ | ✗ | ✗ | ✗ | ✗ | ✗ | ✗ | ✗ | ✓ | ✗ | ✗ | ✗ | ✓ | ✗ | ✓ | ✗ | ✗ | ✗ | ✗ | ✓ |
| *infer representation given an image* | ✓ | ✓ | ✓ | ✓ | ✓ | ✓ | ✓ | ✓ | ✓ | ✗ | ✓ | ✓ | ✗ | ✓ | ✗ | ✓ | ✓ | ✓ | ✓ | ✓ | ✓ |
| *generate new plausible images/scenes* | ✗ | ✗ | ✗ | ✗ | ✗ | ✗ | ✗ | ✓ | ✓ | ✗ | ✗ | ✗ | ✗ | ✓ | ✓ | ✗ | ✗ | ✓ | ✗ | ✗ | ✓ |
| *learns a prior over individual object appearances* | ✓ | ✗ | ✗ | ✓ | ✓ | ✓ | ✓ | ✓ | ✓ | ✓ | ✓ | ✓ | ✗ | ✓ | ✓ | ✓ | ✓ | ✗ | ✗ | ✗ | ✓ |
| *non-learnt 3D rendering process* | ✗ | ✗ | ✗ | ✗ | ✗ | ✗ | ✗ | ✗ | ✗ | ✗ | ✗ | ✗ | ✗ | ✓ | ✗ | ✗ | ✓ | ✓ | ✗ | ✓ | ✓ |

Table 4: Comparison of capabilities of related unsupervised models. Check (✓) - capability supported by the model; cross (✗) - capability not supported.

fair comparison with NeRF-VAE* and uORF*, we provide these methods with the same inductive biases as we used for our method. In particular, (i) we specify possible scene and foreground boundary and use the same number of samples along a ray as described in Implementation Details Sec. 8; (ii) we provide baselines with an equivalent architecture to ours; (iii) we modify NeRF-VAE to take just one view as input, and use non-iterative amortized inference for computational tractability. Furthermore, to make these methods work on the GQN dataset, we have found that it is useful to provide multiple images to the encoder during training. However, during test time, we evaluate all methods on a single image. We hence first pretrain both baseline approaches with multiple views with the same pooling mechanism as described in our method's implementation details. We then interrupt the training and continue training the model with encoder taking $M = 1$ images as input.

## 11 Related work

In Tab. 4, we enumerate various works addressing similar tasks to our proposed approach, and note whether they support various features.