# OpenReview forum: "Unsupervised Causal Generative Understanding of Images"
_NeurIPS.cc/2022/Conference — NeurIPS 2022 Accept_

### Official Review · Reviewer_tfzG · 2022-07-08

**Rating:** 6
**Confidence:** 4
**Soundness:** 2 fair
**Presentation:** 3 good
**Contribution:** 3 good

**Summary:**

This paper presents a slotted generative model with a NeRF rendering decoder. Each slot parameterizes a scene function and the decoder mixes the scene functions while rendering (summing densities and averaging colors). The model is trained as a 2-stage VAE - first training the slotted model VAE and then a "global" VAE which models the marginal posterior distribution of the first model. The slot latents are factorized into "appearance" and "position" and the latter is one-hot encoded (relaxed with Gumbel softmax) and used as a convolution kernel to "shift" the scene function in scene space while rendering.
The model receives one or more input views of a scene and is trained as a VAE, much like NeRF-VAE but with slots.
An inference procedure is also proposed to work in the case of OOD inputs where the amortized encoder is not expected to work.
The method is demonstrated to work nicely on two synthetic datasets - GQN and ARROW.
There are some experiments that show that the model is able to learn factorized representation and these can be manipulated.

**Questions:**

* How is the conditional model p(z_s|z_g) structured? I looked at the supplementary material and it wasn't clear - how is the mapping for a single latent to multiple slots done? is it just an MLP which scales with the number of slots?

* Follow-up to the above: can you change the number of objects without affecting the number parameters in the model? Consequently - can you instantiate a model with a different number of slots than used in training?

* What prevents the object location to be learned in a non-canonical position and compensate with the appropriate shift through the inferred position? There's nothing constraining the object to be learned around the origin.

* Have you tried the actual learned encoder on OOD data instead of using MCMC inference? I know this is claimed not to work in the paper but I didn't see an experiment showing this.
* Follow-up to the above: have you tried amortized iterative inference for evaluation of OOD? (IODINE works on unseen combinations of color and shape, for example)

**Limitations:**

I think the authors addressed the model limitations well in the paper - the above question directly refer to some of the missing points I feel.

**Strengths And Weaknesses:**

Originality:
  While mostly an amalgamation of existing methods this is an interesting model and the fact that it works, even on synthetic data, is non-trivial. The one-hot encoding of position is a nice touch and I suspect makes the model work better.
  I am, however, not convinced by the "Causality" claim of the model - I am not a causality person, but I feel this is a stretch - this model is not more "causal" than IODINE, for example - it does, maybe, reflect the underlying generative process better (through cameras for example) but I feel this is a bit of stretch (see more below)

Quality:
  The paper is quite thorough, and all in all quite good. I feel the choice of comparing only to NeRF based baselines is a weakness - much of the rendering process, especially in simple scenes like these, can be learned well by 2D models (up to NVS) and these should have been compared to (for example, GQN and IODINE). I would want to see analysis of the global latent, and how it affects things (the authors do mention this under "limitations")

Clarity:
  The paper is written nicely and quite clearly, but I feel several elements are not described well - especially the role and structure of the global latent, see below.

Significance:
  All in all I feel this is an important contribution to the community with some non-trivial advances.

---

> ### Author Response · Authors · 2022-08-02
> **Response to reviewer tfzG (1/2)**
>
> Thank you for your valuable suggestions! We are glad that you valued our work’s originality, thoroughness and our proposed encoding of object positions. We note that your major concern is the lack of a 2D baseline and ablation study. First, we'll address the claimed lack of 2D baseline by providing the requested 2D baseline (IODINE); also note that we did already include a strong 2D baseline (Slot-Attention). Second, we’ll add an ablation study on the global latent variable and on MCMC inference. Finally, based on your questions, we'll edit the text, adding details on architecture and explicitly enumerating the benefits of our canonical-space object representation over prior works using spatial mixture models (uORF, ObSuRF).
>
> > I am not convinced by the "Causality" claim of the model - I am not a causality person, but I feel this is a stretch - this model is not more "causal" than IODINE, for example - it does, maybe, reflect the underlying generative process better (through cameras for example) but I feel this is a bit of stretch
>
>
> Per the Stanford Encyclopedia of Philosophy, “A causal model entails the truth value, or the probability, of counterfactual claims about the system; it predicts the effects of interventions; and it entails the probabilistic dependence or independence of variables included in the model”. Our model satisfies these criteria. For example, we can intervene on the distribution of object layouts or positions, without affecting that of object appearances (unlike IODINE). Similarly, our method allows calculating counterfactuals, such as editing the position of an object in a given image, by encoding the image into our latent representation (i.e. calculating the conditional distribution on object positions and appearances given pixels), then intervening on its position only. Notably, our structured causal model achieves this even without requiring interventional data for training. This is a significant step from current “statistical” unsupervised segmentation models which do not define counterfactual or interventional distributions and cannot perform inference in the OOD setting. This is because prior works cannot easily be intervened to account for distribution shift as they are neither generative nor define a probability density over scenes/objects (e.g. Slot-Attention and uORF), or they model the scene with a single latent variable (e.g. NeRF-VAE).
>
> >the choice of comparing only to NeRF based baselines is a weakness - much of the rendering process, especially in simple scenes like these, can be learned well by 2D models (up to NVS) and these should have been compared to
>
> We'll add another 2D baseline – IODINE; we’ll post results from this here as soon as they’re ready. However, note that we already compare to Slot-Attention, which uses a 2D spatial broadcast decoder rather than a NeRF. We chose this precisely to test whether a well-known 2D discriminative compositional baseline can generalize to different out-of-distribution axes. Our experiments show (see Table 3 in the supplementary) that it does not generalize well to novel camera viewpoints, hence demonstrating what we argue in the paper – that a 2D baseline is not able to perform such generalization.
>
> >I would want to see analysis of the global latent, and how it affects things
>
> As requested, we'll include an ablation study on the global latent and explain why it is needed in section 3. Specifically, it is needed to correctly model the probability of entire scenes (i.e. arrangements of objects) - without it, the generative model would incorrectly assume independence of objects (e.g. it would not be able to model object relationships, non-intersection of objects and the fact that objects are not floating in space but are on common floor). In contrast, our model samples plausible scenes, correctly learning non-trivial relationships between objects (e.g. generated scenes for Arrow dataset contain one arrow pointing towards the one odd object).

---

> > ### Author Response · Authors · 2022-08-02
> > **Response to reviewer tfzG (2/2)**
> >
> > >How is the conditional model p(z_s|z_g) structured? I looked at the supplementary material and it wasn't clear - how is the mapping for a single latent to multiple slots done? is it just an MLP which scales with the number of slots?
> >
> > Yes, it is an MLP mapping the Gaussian variable $\mathbf{z}^g$ to a list of object variables. We'll clarify this in the paper. It would be interesting future work to consider a permutation-invariant set decoder.
> >
> > > Follow-up to the above: can you change the number of objects without affecting the number parameters in the model? Consequently - can you instantiate a model with a different number of slots than used in training?
> >
> > Yes, we can instantiate a model with a different number of slots than used in training though it will not allow us to trivially use the learnt global prior. This is orthogonal to our work and could be addressed by using a different prior (e.g. autoregressive) or as in GENESIS-v2 (Engelcke, 2021).
> >
> > >What prevents the object location to be learned in a non-canonical position and compensate with the appropriate shift through the inferred position? There's nothing constraining the object to be learned around the origin.
> >
> > Objects can be learned with an offset from the origin within their canonical reference frame, yes. However, this will result in a corresponding offset in the peak of the Gumbel-Softmax distribution over locations, still allowing consistent rendering of the scene, and learning of contextual relations among objects. Furthermore, note that working in a canonical space still has several advantages over prior mixture models. First, rendering an image with canonical object representation is more efficient as it requires less memory (it only requires rendering NeRF in fixed canonical space) while a non-canonical representation requires sampling points in the full scene volume, which quickly becomes intractable for a large scene. Second, from a representation learning perspective, the representation of an object is ideally invariant to its position, whereas learning objects in non-canonical space wastes model capacity by requiring different representations for each position. Third, from a causality perspective, answering probabilistic and counterfactual queries about object position requires a causal model incorporating position explicitly, along with appropriate conditional distributions.
> >
> >
> > >Have you tried the actual learned encoder on OOD data instead of using MCMC inference? I know this is claimed not to work in the paper but I didn't see an experiment showing this.
> >
> > We agree this is important for rigorous evaluation so we'll include these results in the ablation study. We will post the results of these new experiments here as a comment once they are ready, and will also update the manuscript.
> >
> > >Follow-up to the above: have you tried amortized iterative inference for evaluation of OOD? (IODINE works on unseen combinations of color and shape, for example)
> >
> > IODINE only showed qualitative results in their figure 7 without quantitative evaluation of OOD colors/shapes. Nevertheless, as mentioned, we'll add IODINE as a baseline, to give an example of how a method with iterative amortized inference performs. However, there is a theoretical argument that discriminative models and non-structured generative models cannot be trivially adjusted/intervened to model distribution shift, hence performing inference on OOD data that should have a zero probability is not mathematically sound.

---

> > > ### Comment · Reviewer_tfzG · 2022-08-04
> > > **Thank you for the responses for my concerns**
> > >
> > > All in all I would say your responses have answered most of my concerns.
> > >
> > > I realize the global latent is the one responsible for modeling global scene structure - by "analysis" I meant more trying to understand *how* it does that - for example - a simple latent traversal may shed some light on what structures this latent holds.
> > >
> > > Nevertheless - assuming the IODINE baseline would be included in the paper and comparison, as well as the respective discussion I would say I am leaning more towards accepting the paper (which strengths I mentioned in my original review).

---

> > > > ### Author Response · Authors · 2022-08-09
> > > > **Results for experiments requested by Reviewer tfzG**
> > > >
> > > > Thank you once again for your review, we are glad to hear we have resolved your concerns.
> > > >
> > > > > by "analysis" I meant more trying to understand how it does that - for example - a simple latent traversal may shed some light on what structures this latent holds.
> > > >
> > > > Thanks for the suggestion, we agree this would be informative and will add latent space traversals to the camera-ready version of the manuscript.
> > > >
> > > > We have now run the following additional experiments that you requested:
> > > >
> > > > 1. IODINE as an additional 2D baseline with iterative amortized inference
> > > > 2. ablation experiments on our model with...
> > > >    - encoder (amortized inference) replacing MCMC inference in our model on OOD data
> > > >    - removing the high-level prior $p_{\theta}(\mathbf{z}^s\mid\mathbf{z}^g)$
> > > >
> > > > In addition, following the request of Reviewer uNLg, we now also report the performance of Slot-Attention using our MCMC inference scheme instead of amortized inference.
> > > >
> > > > In the top-level comment (named "Requested Experiments"), we give a summary of these results; for full details, please see the updated manuscript.
> > > >
> > > > *We thank you again for engaging in the discussion, and for your valuable suggestions!*

---

### Official Review · Reviewer_gUkP · 2022-07-12

**Rating:** 7
**Confidence:** 5
**Soundness:** 4 excellent
**Presentation:** 3 good
**Contribution:** 3 good

**Summary:**

This paper proposes a generative model of images that factors appearance of the image into background and multiple objects — each object is associated with its own NERF based appearance model drawn from a global prior and together can be used to forward render an associated image in a differentiable way.  Latent object appearance models are inferred using a VAE-like framework with neural-net encoders.

Finally by replacing certain learned priors by uniform priors, the authors show that their models are able to work in OOD settings quite accurately.  Results in this paper are all on synthetic baselines.



**Questions:**

* I cannot figure out how are shape and color separated in this model — it’s possible that I am missing something, but shape and color variables seem to be treated symmetrically throughout, so what forces one variable to map to shape and the other to map to color?
* Related to this question, it would be useful to provide further factorization in Equation 2 to explain the joint distribution of shape, color and position of a single given object.
* I wonder if there is a principled way to decide when it is okay to replace a learned prior with a uniform distribution.  In other words, when is it okay to extrapolate from the dataset?  Clearly the answer cannot be “all the time” — could the authors shed some light on how to make this decision appropriately?


**Limitations:**

yes.

**Strengths And Weaknesses:**

Strengths:

A full instance-level generative model of image appearance is arguably one of the “holy grails” of computer vision — and the fact that this paper takes a step in this very ambitious direction and achieves reasonably good results (on synthetic data) is a strength.

There are two possible families of comparison — NERF based models which generally do not have a probabilistic object based decomposition of the scene, and 2d models of image composition which do capture object based decompositions, but are not able to learn from multi-view data in the same way. Compared to other works along the same lines (e.g. slot attention), the authors also show strong performance in their model’s ability to generalize to out-of-distribution data.

Another strong point of the paper is the attention paid to reliable optimization.  For example, the authors find that modeling positions using a categorical variable helps (compared, e.g, to using naive spatial transformer models which would model pose using a continuous variable).  The idea is similar to the way that Faster R-CNN works for object detection --- use a discrete grid to predict at an "anchor grid" of positions, then use ROIAlign (which is basically an axis aligned STN) to crop features.


Weaknesses:

There are some major things that are not modeled in this work but would be important in the long run — these include orientations (or a more general space of transformations beyond point-to-point translation) and semantics (there is not a way that the concept of “chair” can be learned by training on many images).  Also, even though images can be “out-of-distribution” in the sense of having different numbers of objects, being placed in different locations or having novel compositions, I don’t believe that generalizing to novel types of objects is handled well in this approach.

Another weakness of the approach (mostly having to do with the fact that it requires MCMC) is the running time, which the authors acknowledge to be a limitation.  I would recommend mentioning more specifics about time requirements.

---

> ### Author Response · Authors · 2022-08-02
> **Response to reviewer gUkP (1/2)**
>
> Thank you for the encouraging review! We are glad you found our work to be an ambitious step towards an instance-level generative model of scenes – which we agree with you is a “holy grail” of computer vision. As you request, we’ll clarify the factorisation of the probabilistic model and detail how this is implemented. We'll also add an explanation of how the NeRF architecture for appearance representation facilitates disentanglement of shape and color. Lastly we'll include more details on the MCMC inference scheme and its computational efficiency.
>
> >I cannot figure out how are shape and color separated in this model — it’s possible that I am missing something, but shape and color variables seem to be treated symmetrically throughout, so what forces one variable to map to shape and the other to map to color?
>
>
> We achieve disentanglement (one variable to map to shape and one to color) by allowing only the shape variable to influence the density (i.e. opacity) of 3D points. Specifically, an object's NeRF is made out of two neural networks: the first takes as input a 3D point and the object’s shape variable and outputs a 1-dimensional opacity (implicitly defining the 3D shape) and an embedding h; a second network takes the embedding h and the object’s appearance variable and outputs an RGB color. Hence, opacity only depends on the latent shape variable, not color. This was mentioned in section 6 of the supplementary, but we'll also add a brief note in the main text.
>
> >Related to this question, it would be useful to provide further factorization in Equation 2 to explain the joint distribution of shape, color and position of a single given object.
>
>
> Thank you for pointing this out. As requested, we'll give the factorization of shape, color and position (which are in fact conditionally independent given $\mathbf{z}^g$), in Equation 2.
>
> >I wonder if there is a principled way to decide when it is okay to replace a learned prior with a uniform distribution. In other words, when is it okay to extrapolate from the dataset? Clearly the answer cannot be “all the time” — could the authors shed some light on how to make this decision appropriately?
>
>
> There are two possible interpretations to this question.
>
> The first is: how to determine whether or not a given variable (in some OOD test setting) is in fact drawn from a distribution than that seen during training? One approach is to check whether the probability of the observed variable is below some predetermined threshold under the non-intervened (training) distribution. Some work on novelty detection has explored similar ideas in the past, e.g. the classic “Novelty detection and neural network validation” (C. Bishop, 1994), and more recent works in the generative model literature on OOD detection (e.g. “Likelihood Ratios for Out-of-Distribution Detection”, J. Rien et al., NeurIPS 2019). It would be interesting future work to investigate how such techniques may be combined with our approach.
>
> The second is: how to determine which variables are suitable to have their priors replaced by a uniform distribution in particular? This is an interesting question, and in general depends which mechanism (i.e. conditional) is intervened on. In the case of our scene-level prior $p_{\theta}(\mathbf{z}^s|\mathbf{z}^g)$, it’s appropriate to replace it with a uniform distribution when we know that similar types of objects are going to be seen as during training, and we believe that there is no constraint on where (and in what combinations / compositions) those objects will appear.

---

> > ### Author Response · Authors · 2022-08-02
> > **Response to reviewer gUkP (2/2)**
> >
> >
> > >There are some major things that are not modelled in this work but would be important in the long run — these include orientations (or a more general space of transformations beyond point-to-point translation) and semantics (there is not a way that the concept of “chair” can be learned by training on many images).
> >
> > We agree that there are other important latent factors that could be incorporated explicitly in our approach, such as orientations and object classes. Hopefully future works will consider these extensions, including in the OOD setting where they are subject to distribution shifts.
> >
> > >Another weakness of the approach (mostly having to do with the fact that it requires MCMC) is the running time, which the authors acknowledge to be a limitation. I would recommend mentioning more specifics about time requirements.
> >
> >
> > Thank you for your suggestion, we'll provide full details in section 7. We run MCMC until convergence or until 15K iterations are reached. However, on the same note, we want to highlight that our structured causal model allows MH proposals that affect only one object while keeping other variables fixed. This increases efficiency in contrast to MCMC on non-structured models as it allows caching computation and only re-rendering parts of the scene that need to be considered for a proposed change (e.g. just a single object). In contrast, MCMC on non-structured models must render the scene from scratch. Moreover, each MH step need not revert any progress made on other variables: e.g. if the background is perfectly inferred but objects are not, then the MH steps can modify objects but leave the background intact.

---

### Official Review · Reviewer_KD5e · 2022-07-12

**Rating:** 6
**Confidence:** 3
**Soundness:** 3 good
**Presentation:** 3 good
**Contribution:** 3 good

**Summary:**

The authors proposed a VAE-based NERF-like object-centric framework to model 3D-scenes. The model represents objects as separate NERFs. Moreover, authors make architectural choices that reflect causal independent mechanisms, so this enables OOD generalization. Plus, a novel MCMC inference scheme is proposed to infer out-of-distribution scenes. The model outperforms other models in OOD setting and performs competitively in In-distribution set-up.

**Questions:**

Ablation study questions

Main questions
1) Can you explicitly state all architectural choices you made to enable independent mechanisms and perform an ablation study on these choices?
2) Do you need a two-step generation? The general latent layout of the scene is generated firstly and then latents of objects are generated based on this layout. Can you ablate it?
3) Did you apply a new MCMC procedure to your baselines (slot-attn, uORF, NERF-VAE) for OOD setup? If not -- why?
4) Can you ablate your MCMC procedure for OOD setup and show what scores the model obtains when one uses an encoder to obtain latents?

Small issues
1) I did not understand figure 1, can you maybe elaborate on this picture a little bit more.
2) I suggest to the author to include a graphical model in the main text to increase readability.


**Limitations:**

-

**Strengths And Weaknesses:**

Strengths

- the idea and necessity of a novel MCMC inference scheme are intuitive and well-explained
- the model outperforms baselines in OOD setting and performs competitively in In-distribution set-up.
- the model is OC and generative (that is novel as far as I understand) compare to its OC counterpart (uORF), so I believe it’s easier to edit a scene object-wise.
- authors make architectural choices to reflect the causal independent mechanisms.

Weaknesses

- there is no ablation study on model design -- this is my main concern
- the architectural choices that make the model more causal are not explicitly stated in one place in the main text, so it’s hard to follow what exactly these choices are. And there is no ablation study on these choices either.

Overall, in my opinion, the authors implemented the right priors to make OOD generalization possible and the paper seems novel to me.

I am willing to increase my score if a proper ablation study will be performed.

---

> ### Author Response · Authors · 2022-08-02
> **Response to reviewer KD5e (1/2)**
>
> Thank you for your review! We are glad that you found our work to be a significant step from recent work (uORF, NeRF-VAE) and appreciated ​​the necessity of a novel MCMC inference scheme for OOD image understanding. We note that your main concern is the lack of ablation experiments on model design – to mitigate this, we'll provide ablation experiments on important aspects of the method (compositionality, inference scheme and high-level prior over objects). We will also make the suggested small edits to the manuscript, including drawing the graphical model and clarifying figures.
>
>
> >there is no ablation study on model design -- this is my main concern. I am willing to increase my score if a proper ablation study will be performed.
>
> To fulfil this request, we will include an ablation study on several important aspects of our method. (1) We will ablate our MCMC scheme, replacing it by amortised inference with an encoder. (2) We will ablate compositionality of our method, instead modelling the scene with a single latent vector. (3) We will remove the high-level prior over objects, and evaluate samples generated by the resulting model. These experiments are currently running, and we’ll post results here once they’re completed.
>
> Regarding the second ablation, we also want to highlight that our structured causal model allows MCMC proposals that affect only one object (while still considering the probability of the entire scene and image). This increases efficiency versus MCMC on non-structured models. First, it allows caching computation and only re-rendering parts of the scene that need to be considered for a proposed change (e.g. just background). In contrast, MCMC on non-structured models renders the scene from scratch. Second, each MH step does not revert any progress made on other variables: e.g. if the background is perfectly inferred but objects are not, then an MH proposal may change only an object, leaving the background intact.
>
> >the architectural choices that make the model more causal are not explicitly stated in one place in the main text, so it’s hard to follow what exactly these choices are. And there is no ablation study on these choices either. Can you explicitly state all architectural choices you made to enable independent mechanisms and perform an ablation study on these choices?
>
> Thanks for the suggestion – we'll summarise them in the introduction, and provide an ablation study to evaluate their importance (see above). In short, we require that the generative model reflects the physical process by which images arise (3D objects are placed into a scene, and light rays reflected by them arrive at a camera with some particular viewpoint). The important aspects are:
> 1. a non-learnt rendering mechanism – this is guaranteed to generalise correctly to OOD data; it contrasts with prior works that learn the rendering process (e.g. using a CNN that inputs rendered features and outputs the image; e.g. GIRAFFE, Niemeyer, CVPR 2021)
> 2. an explicit disentangled representation of the causal variables (e.g. object shapes and positions) – this allows us to perform interventions and counterfactual causal inference (e.g. conditioning on an image to infer the corresponding latents, then manipulating object positions); this contrasts with works that model the scene without disentangling the proper causal variables (object positions, layouts, appearances, etc. – e.g. spatial mixture models).
> 3. separation of the mechanisms (conditionals) for per-object appearances and for scene compositions (layouts), so the latter can be intervened on without affecting the former; this contrasts with methods that have a single global decoder (e.g. NeRF-VAE).
>
> >Do you need a two-step generation? The general latent layout of the scene is generated firstly and then latents of objects are generated based on this layout. Can you ablate it?
>
> Yes, we do need two-step generation to correctly model the probability distribution over entire scenes (i.e. compositions of objects) – without it, the generative model would incorrectly assume independence of components z^s (e.g. it would not be able to model object relationships, non-intersection of objects and the fact that objects are not floating in space but are on common floor). In contrast, our model samples plausible scenes, correctly learning non-trivial relationships between objects (e.g. generated scenes contain one arrow pointing to one odd object on the Arrow dataset).
>
> To demonstrate this, we’ll fulfil the request to provide an ablation experiment on the high-level prior, and we'll include the results in Table 2.

---

> > ### Author Response · Authors · 2022-08-02
> > **Response to reviewer KD5e (2/2)**
> >
> > >Did you apply a new MCMC procedure to your baselines (slot-attn, uORF, NERF-VAE) for OOD setup? If not -- why?
> >
> > We agree this is important for rigorous evaluation; so we'll provide an ablation with MCMC on an non-structured generative model. Though we note that it is not possible to apply MCMC to other baselines (Slot-Attention, uORF) as they are non-probabilistic and don’t define a probability density over scenes/objects.
> >
> >
> > >Can you ablate your MCMC procedure for OOD setup and show what scores the model obtains when one uses an encoder to obtain latents?
> >
> > Yes, we’ll provide these results in Table 2.
> >
> > >I did not understand figure 1, can you maybe elaborate on this picture a little bit more.
> >
> > We'll update the caption to be clearer. The aim of figure 1 is to visualise that in contrast to statistical models which represent one distribution, a causal model represents (and generalises to) many different distributions, because it can be intervened on to model a different density over scenes. We show that our model accurately infers scene representation for various OOD images as shown in the figure. In contrast, prior statistical models only support inference on the I.I.D. data but cannot be intervened to generalize to OOD images that have zero probability in the training set.
> >
> > >I suggest to the author to include a graphical model in the main text to increase readability.
> >
> > Thanks for the suggestion – we’ll add this in section 3.

---

> ### Author Response · Authors · 2022-08-09
> **Results for experiments requested by Reviewer KD5e**
>
>
> We have now run the following additional experiments you requested, to examine the benefit of different aspects of our approach:
>
> 1. ablation experiments on our model with...
>    - encoder (amortized inference) replacing MCMC inference
>    - unstructured full-scene latent representation replacing per-object structure in the generative model
>    - removing the high-level prior $p_{\theta}(\mathbf{z}^s\mid\mathbf{z}^g)$
> 2. Slot Attention baseline with MCMC inference instead of amortized inference
>
> In addition, following the request of Reviewer tfzG, we have added IODINE as a further 2D baseline.
>
> In the top-level comment (named "Requested Experiments"), we give a summary of these results; for full details, please see the updated manuscript.
> We hope this now addresses your concern regarding lack of ablation study and rigorous analysis of how MCMC helps the model; we kindly ask that you consider raising your score as promised in your review.

---

### Official Review · Reviewer_uNLg · 2022-07-13

**Rating:** 6
**Confidence:** 5
**Soundness:** 2 fair
**Presentation:** 3 good
**Contribution:** 2 fair

**Summary:**

This paper proposes a generative model for unsupervised object-centric 3D scene understanding. The proposed model is claimed as "causal" as it is baked in with a multi-object Neural Radiance Field with explicit latent variables for color, shape, and object positions. This model is trained with ELBO in Variational Bayes. To demonstrate that the introduction of the "causal" structure can help generalize the inference of latent variables to OOD scenes, i.e. scenes not covered in the support set of the training data, the authors further propose an MCMC sampling method for test time only. The proposed model shows superior performance in GQN dataset where objects are iid and ARROW datasets where there is a specific correlation between objects.

**Questions:**

Apart from the questions I raised in the weakness part above, I would appreciate it if the authors would like to answer the following:

1. Since the way of inference in OOD images is different in the proposed model and existing models (MCMC vs Variational Encoder), it is unclear which component between the explicit generative model and the MCMC inference plays a more critical role in the success of OOD generalization. Maybe the authors would like to apply their MCMC samplers to the trained decoder of Slot Attention, uORF, and NeRF-VAE? I believe this ablation can be beneficial for readers who are interested in OOD generalization.
2. If time allows, authors may also want to compare these models in datasets where there are more authentic textures, for example, the CLEVR-TEX dataset?
3. Authors claimed to use uniform distribution to replace the learned prior when generalizing to OOD images, how is that implemented and what is the rationale here? More specifically, what is the range of the support space, and why so?

I will be happy to raise my rating if authors would like to resolve some of my concerns.

--------------------

Post discussion: Thanks authors for providing results on experiments I requested, I have raised my rating.

**Limitations:**

Apart from the limitation authors listed in Sec 4.5, I believe the authors may also want to talk about how statistical assumptions in the proposal distribution in MH sampling may affect the OOD inference, And how can the assumption of knowing there is indeed a distribution shift can be fulfilled in reality.

**Strengths And Weaknesses:**

\+ This paper is very well written. I liked the narrative in which the authors first describe the representation and the generative models with probabilistic language, then introduce the inductive bias of NeRF, and then move on to introduce training and inference. This makes the method section very easy to follow.

\+ The introduction of the scene variable is fairly interesting. Existing methods for object disentanglement mostly assume iid prior over objects. The hierarchical design of the generative model echoes the "conditional independence" in causality, making the assumption in models more generic.

\+ Though couples of existing works use NeRF as a scene generator, the explicit modeling of the latent variable and the MCMC sampling in OOD scenes are novel contributions. In particular, the observation that the variational encoder from Variational Bayes training is the critical bottleneck for OOD generalization is very sharp.

\+ As far as I know, generalizing to novel viewpoints is something not covered by the existing literature on multi-object representation learning.



\- Though I liked the idea of using MCMC sampling for inference in OOD images, I have some concerns about its concrete realization. First, to account for the distribution shift in the latent causal factors, the proposed method needs to manually replace the learned prior of latent variables with uninformative distributions. Then is it fair to compare with other existing models since here the model is actually informed of which causal factor is intervened in the distribution shift? Second, the combination of Langevin dynamics and Metropolis-Hastings is interesting, but why the proposal of each object can be "considered" independently?

\- The comparison with the baselines is not completely fair. First, Slot Attention model does not use either the viewpoint information or multiple views of the same scene. Maybe authors should try a variant of it imitating how GQN extends VAE? I am also curious why the trained Slot Attention model does not generalize to objects with OOD number of objects since it should be successful according to the original paper. Second, NeRF-VAE is not encouraged to have structured latent representation, so we kind of know a priori it will fail the composition test, maybe the authors should try some disentangled VAEs such as beta-VAE? Third, the correlation in the ARROW dataset that the arrow object is always pointing to a special object is known a priori and the proposed model seems to perform well in this dataset by incorporating this knowledge into the model. In reality, we don't really know what kind of correlation there are among objects, how would the proposed model still be generic?

---

> ### Author Response · Authors · 2022-08-02
> **Response to reviewer uNLg (1/2)**
>
> Thanks for your review! We are glad that you found our work to be well-written, complimented our novel MCMC scheme (as a possible solution for amortised inference being a critical bottleneck for OOD generalization), and valued the structured graphical model that facilitates MCMC. First, we address your concern about comparison to baselines, including adding some new experiments. Second, we’ll provide the requested ablation studies (and more). Finally, we clarify that our model learns object relationships without any supervision, since we believe you misunderstood that our approach cannot do so.
>
> >is it fair to compare with other existing models?
>
>
> Yes, we believe so, as no related work can in theory account for the distribution shift. In detail, Slot-Attention and uORF are discriminative approaches that do not trivially support accounting for distribution shifts, while NeRF-VAE models the scene with one latent variable which entangles true causal variables (such as object positions, relationships between components, etc.) and cannot be intervened on to model a different distribution of scenes. In contrast to prior work, our structured explicit generative model can be intervened on, can compute counterfactuals (e.g. calculating conditional distributions on object positions and appearances given an image, then modifying some of these variables), and can perform out-of-distribution inference.
>
> However, given the question arose to the reviewer, we’ll also do the following:
> * To strengthen the empirical validation, we’ll add an ablation study on aspects of our model design.
> * We’ll emphasise in the experiments and related work section that prior related work cannot account for distribution shifts.
> * We’ll clarify in the introduction how our model facilitates intervention and counterfactuals.
>
>
> >but why the proposal of each object can be "considered" independently?
>
> We meant that our structured causal model allows Metropolis-Hastings proposals that affect only one object while keeping other variables fixed. It still is a sound MCMC scheme which considers the joint probability of the entire image and latent scene. This approach increases efficiency in contrast to MCMC on non-structured models for two reasons. First, it allows caching computation and only re-rendering parts of the scene that need to be considered for a proposed change (e.g. just background). In contrast, MCMC on non-structured models renders the scene from scratch. Second, each MH step does not revert any progress made on other variables: e.g. if the background is perfectly inferred but objects are not, then an MH proposal may change only an object, leaving the background intact.
>
> Fixes:
> * We’ll clarify what me meant by “considered independently”
> * We’ll explicitly describe the efficiency of our MCMC scheme on our structured generative model in section 3.
>
>
> >Slot Attention model does not use either the viewpoint information or multiple views of the same scene. Maybe authors should try a variant of it imitating how GQN extends VAE? I am also curious why the trained Slot Attention model does not generalize to objects with OOD number of objects
>
> uORF (which we already compare to as a baseline) is one possible 3D extension of Slot-Attention; hence, we do not feel it is necessary to add another one. We chose Slot-Attention precisely to test whether a standard 2D discriminative compositional baseline can generalize to different OOD settings: Table 3 in the supplementary shows that it cannot generalize to OOD viewpoints, but it is among the best on generalization to OOD number of objects (consistent with the Slot-Attention paper).
>
> Fixes:
> * We’ll clarify this in section 4.
> * We’ll add more analysis based on results in Table 3 in the Supplement.
>
>
> >maybe the authors should try some disentangled VAEs such as beta-VAE?
>
> Fixes:
> * We'll provide results using beta-VAE.

---

> > ### Author Response · Authors · 2022-08-02
> > **Response to reviewer uNLg (2/2)**
> >
> > >the proposed model seems to perform well in this dataset by incorporating this knowledge into the model
> >
> > We believe you slightly misunderstood and underestimated our method, by assuming that it requires ground-truth relationships between objects (hence is not fully unsupervised). However, our work is truly unsupervised – it does not require any ground-truth relationships or any labels, and we explicitly state this throughout the manuscript (L14, L34, L142). Our model learns relationships  (e.g. the fact that the arrow always points at the odd object) automatically without supervision with the hierarchical prior described in Equation 2, and it successfully samples plausible scenes (Table 2 and Figure 3).
> >
> > Additionally, we’ll perform an ablation experiment where the prior in Equation 2 is removed.
> >
> > >Maybe the authors would like to apply their MCMC samplers to the trained decoder of Slot Attention, uORF, and NeRF-VAE?
> >
> > Fixes:
> > * We’ll provide the requested experiment with MCMC on an unstructured generative model. Though note MCMC cannot be applied to the other baselines (Slot Attention, uORF) as they are non-probabilistic and don’t define a probability density over scenes/objects.
> > * We’ll provide extra experiments for our model by ablating MCMC with amortised inference.
> >
> > >Authors claimed to use uniform distribution to replace the learned prior when generalizing to OOD images, how is that implemented and what is the rationale here? More specifically, what is the range of the support space, and why so?
> >
> > We use uninformative prior distributions: uniform categorical for location and improper uniform (over the reals, with infinite support) for other variables (shape and appearance embeddings). Rationale: in the OOD setting, we only know that the mechanism describing scene-level relations among objects was intervened on but not what the intervention is; hence, we replace the previously learnt scene prior with a distribution that does not impose any restrictions on possible layouts / compositions of objects.
> >
> >
> > >authors may also want to talk about how statistical assumptions in the proposal distribution in MH sampling may affect the OOD inference
> >
> > Our proposal distribution covers all components latents (i.e. possible object/background appearance embeddings) that were seen in any scene during training. It expresses the assumption that the OOD scene contains known objects but in novel compositions / positions.
> >
> > Fixes:
> > * We’ll include this explanation in section 3.
> >
> >
> > >how can the assumption of knowing there is indeed a distribution shift can be fulfilled in reality
> >
> >
> > A principled way to detect the distribution shift (and therefore to choose to intervene on the variable) is to measure whether the probability of an observation is below some predetermined threshold under the non-intervened distribution. Some work on novelty detection has explored similar ideas in the past, e.g. the classic “Novelty detection and neural network validation” (C. Bishop, 1994), and more recent works in the generative model literature on OOD detection (e.g. “Likelihood Ratios for Out-of-Distribution Detection”, J. Rien et al., 2019). It would be interesting future work to investigate how such techniques may be combined with our approach.
> >
> > > model shows superior performance in GQN dataset where objects are iid
> >
> > Just as in Arrow, GQN objects are non-IID - they have relationships as described in Supplementary (e.g. next to an ‘odd’ wall with color different to the others). Hence, both datasets require a high-level scene prior to model correctly.

---

> > > ### Comment · Reviewer_uNLg · 2022-08-03
> > > **Clarifying unanswered questions**
> > >
> > > Thanks authors for the detailed reply. However, I find some major questions in my review unanswered. To help better understand my concerns, here are some clarifications:
> > >
> > > > object relationships
> > >
> > > I didn't mean the model knows explicitly what the relation is. I was just curious if the relation in ARROW is too particular for the designed independence assumption. What if the relation is the augmented with "the attribute of color and shape are correlated instead of independent"? Would the shape and the color variable still be able to be disentangled?
> > >
> > > > mcmc on other existing models for ablation study
> > >
> > > Both Slot Attention and uORF learn a mixture decoder, which can easily parametrize a pixel-wise Gaussian generator with a small sigma. As for the prior, authors may consider the proposed idea of using uninformative prior, which can make the ablation more specific (the same MCMC inference with the same OOD handling on different generative models). This will help answer if the generative model design or the MCMC inference contributes more to OOD performance.
> > >
> > > > knowledge of distribution shift
> > >
> > > In OOD generalization, how does the model know which latent variable experience a distribution shift? In the reported experiments, do you manually pick latent variables to change priors to uninformative ones based on your knowledge of the different types of OOD?

---

> > > > ### Author Response · Authors · 2022-08-03
> > > > **Clarifying unanswered questions (1/2)**
> > > >
> > > > Thanks for the quick response and clarifications!
> > > >
> > > > >I was just curious if the relation in ARROW is too particular for the designed independence assumption. What if the relation is the augmented with "the attribute of color and shape are correlated instead of independent"?
> > > >
> > > > We emphasise that our model does **not** assume independence among the different scene variables constituting $\mathbf{z}^s$; it only assumes *conditional* independence given $\mathbf{z}^g$. Relationships between these variables are modelled by $p_{\theta}(\mathbf{z}^s|\mathbf{z}^g)$, which can in principle model any relationship among object locations, shapes and colors. Naturally this includes the relation in the ARROW dataset, and would also include the color/shape correlation you mention. It also includes more complex correlations in the GQN dataset described in sec. 8.1 of the supplementary (e.g. certain shapes and colors of object always appear near certain walls).
> > > >
> > > > One possible source of confusion is the reader may assume that in Equation 2, $p_\theta(\mathbf{z}^{shape}\_{i} , \mathbf{z}^{col}\_{i} , \mathbf{z}^{pos}\_{i} | \mathbf{z}^g)$ models the same distribution for **all** objects $i$. This is not the case – in our model each $p_\theta(\mathbf{z}^{shape}\_{i} , \mathbf{z}^{col}\_{i} , \mathbf{z}^{pos}\_{i} | \mathbf{z}^g)$ is a different distribution for each object $i$. These distributions are controlled by a neural network (parameterised by $\theta$), which takes as input the high-level variable $\mathbf{z}^g$, and maps it to separate means for each object’s latent variables (just like a VAE decoder takes the latent variable and maps it to all pixel means). Similar notation is consistently used for such models in the generative modelling literature. However, given that it is important for readers to understand how our model can learn relationships between objects, we will clarify this explicitly in the text. We hope this addresses your concern – assuming independence would indeed be a strong limitation, but this is not the case in our work!
> > > >
> > > > >Would the shape and the color variable still be able to be disentangled?
> > > >
> > > > Our NeRF architecture is designed to ensure object colors and shapes are disentangled. We achieve this by allowing only the shape variable to influence the density (i.e. opacity) of 3D points. Specifically, an object's NeRF is made out of two neural networks: the first takes as input a 3D point and the object’s shape variable and outputs a 1-dimensional opacity (implicitly defining the 3D shape) and an embedding $\mathbf{h}$; a second network takes the embedding $\mathbf{h}$ and the object’s appearance variable and outputs an RGB color. Hence, opacity only depends on the latent shape variable, not color; this also allows changing object color with latent appearance variable without changing object’s 3D shape. This architecture is discussed in section 6 of the supplementary, but we'll also add a brief note in the main text. We emphasise that the fact that these variables are disentangled does *not* preclude them from being correlated via the high-level prior. By disentanglement we mean that modifying either shape or color affects only the corresponding aspect of the generated image; this can still be true even when they have a dependency through a common ancestor variable ($\mathbf{z}^g$ in our case).

---

> > > > > ### Author Response · Authors · 2022-08-03
> > > > > **Clarifying unanswered questions (2/2)**
> > > > >
> > > > > > Both Slot Attention and uORF learn a mixture decoder…  This will help answer if the generative model design or the MCMC inference contributes more to OOD performance.
> > > > >
> > > > > While it is theoretically possible to extend these two existing discriminative baselines to be generative, and to apply our novel MCMC inference scheme on the resulting models, this is a substantial research project in itself, and does not justify criticism of our model. We kindly ask you to reconsider whether it is fair to penalise our work for not comparing with non-trivial and hypothetical extensions of previous approaches. *Please let us know* if you still request extension of baselines, we will try to provide these results before the deadline. Meanwhile, we are running ablation experiments for our own model, to demonstrate the benefit of each aspect (e.g. MCMC vs. amortised inference); we’ll post the results here in the next few days.
> > > > >
> > > > > The fact that other approaches could be developed and compared with ours (e.g. a hypothetical extension of uORF that is generative and incorporates a scene-level prior, and uses our proposed MCMC inference scheme) does not diminish our key contributions – that is, a novel MCMC scheme, a novel causal model of images allowing interventions, counterfactuals and mathematically-principled OOD inference, and empirical results that significantly out-perform existing methods.
> > > > >
> > > > > > In OOD generalization, how does the model know which latent variable experience a distribution shift? In the reported experiments, do you manually pick latent variables to change priors to uninformative ones based on your knowledge of the different types of OOD?
> > > > >
> > > > > Yes, we assume it is known which variable is subject to a distribution shift, but not what the updated distribution is. As mentioned, in future work this could be achieved via the generative model itself – a principled way to detect distribution shift is to measure whether the probability of an observation is below some predetermined threshold under the non-intervened distribution. However, this is beyond the scope of the present work (and far beyond existing works on unsupervised segmentation). As other reviewers noted, ours is the first work to take a step in this exciting direction.

---

> > > > > > ### Comment · Reviewer_uNLg · 2022-08-03
> > > > > > **Further clarification**
> > > > > >
> > > > > > Thanks for your response. I have increased the rating since some of my concerns have been addressed. Here are some follow-ups:
> > > > > >
> > > > > > > each $p_\theta(\mathbf{z}^{shape}_{i} , \mathbf{z}^{col}_{i} , \mathbf{z}^{pos}_{i} | \mathbf{z}^g)$$ is a different distribution for each object $i$
> > > > > >
> > > > > > This is interesting since it really touches upon my concerns about some (conditional) i.i.d. assumptions in the model. However, I hope it would not be too troublesome for the authors to explain further how this difference is achieved and what happens when there are more objects than the maximum number in the training data.
> > > > > >
> > > > > > > MCMC on existing methods
> > > > > >
> > > > > > To clarify, I didn't suggest training a generative model via extending the existing methods to be completely probabilistic. I hope there is an ablation that runs MCMC inference on the **learned** generator/decoder from either Slot Attention or uORF. From what I read, the proposed MCMC inference is somewhat general and should be directly applicable to any slot-based generators. Please correct me if I misunderstood, thank you!

---

> > > > > > > ### Author Response · Authors · 2022-08-05
> > > > > > > **Response to reviewer uNLg: Further Clarifications**
> > > > > > >
> > > > > > > We are glad to hear we have found the source of confusion and clarified that our model can learn complex relationships among object positions and appearances! We will add a further explanation in the paper.
> > > > > > >
> > > > > > > >However, I hope it would not be too troublesome for the authors to explain further how this difference is achieved and what happens when there are more objects than the maximum number in the training data.
> > > > > > >
> > > > > > > This is achieved by the neural network (parameterised by $\theta$) that takes as input the high-level variable $\mathbf{z}^g$, and outputs parameters for all latent variables $\mathbf{z}^s$ (representing object appearances, positions, etc.). This network (whose outputs are denoted $\zeta_\theta(\mathbf{z}^g)$ and $\xi_\theta(\mathbf{z}^g)$ in sec. 3.1 of the paper) is fully-connected (see sec. 6 of the supplementary for the architecture). This allows outputting non-identical distributions for each object’s latents, and capturing relationships between them. Note that this model does not define the probability of latents in the OOD setting with more objects present than during training. Indeed, this is impossible in general – e.g. if during training we always see four objects arranged in a square, then it is not clear what the allowable positions might be for a fifth object.
> > > > > > >
> > > > > > > >To clarify, I didn't suggest training a generative model via extending the existing methods to be completely probabilistic. I hope there is an ablation that runs MCMC inference on the learned generator/decoder from either Slot Attention or uORF
> > > > > > >
> > > > > > > Thank you for the clarification! We will run MCMC inference on the pretrained Slot-Attention model as you suggest. We’ll post results here as soon as this is complete. To further answer your original question of *“[whether] the generative model design or the MCMC inference contributes more to OOD performance”*, we will also provide results from our model but with amortised inference instead of MCMC, to explicitly assess the benefit of the latter. Would this (in addition to beta-VAE and IODINE baselines) upgrade your review from “limited evaluation” to “no major concerns with respect to evaluation”?

---

> > > > > > > ### Author Response · Authors · 2022-08-09
> > > > > > > **Requested experiments for Reviewer uNLg**
> > > > > > >
> > > > > > > We have now run the following additional experiments you requested:
> > > > > > >
> > > > > > > 1. Slot Attention with MCMC inference
> > > > > > > 2. beta-VAE baseline
> > > > > > > 3. ablation experiments on our model with...
> > > > > > >    - encoder (amortized inference) replacing MCMC inference
> > > > > > >    - unstructured full-scene latent representation replacing per-object structure in the generative model
> > > > > > >    - removing the high-level prior $p_{\theta}(\mathbf{z}^s\mid\mathbf{z}^g)$
> > > > > > >
> > > > > > > In addition, following the request of Reviewer tfzG, we have added IODINE as a further 2D baseline.
> > > > > > >
> > > > > > > In the top-level comment (named "Requested Experiments"), we give a summary of these results; for full details, please see the updated manuscript.
> > > > > > > We hope this now addresses all your concerns regarding evaluation; we kindly ask that you consider raising your score if so.
> > > > > > >
> > > > > > > *We thank you again for engaging in the discussion, and for your valuable suggestions!*

---

### Author Response · Authors · 2022-08-02
**Common response and updates for revision**

We thank the reviewers for their detailed feedback! We are glad that all reviewers found our work to be high-quality, well-written and of high significance. Reviewers complimented it as the first explicit step towards unsupervised out-of-distribution scene understanding, and appreciated the original technical contributions, including an efficient MCMC inference scheme and one-hot representation for 3D object positions. The only major concern raised by reviewers [uNLg, KD5e, tfzG] is the lack of an ablation study. To satisfy this reviewers’ request, we’ll provide the following:

* [uNLg, KD5e, tfzG]: we’ll include ablation studies on main aspects of the model design: 1) inference scheme (MCMC vs amortised inference); 2) compositionality (MCMC on structured vs unstructured generative model); 3) high-level prior $p_{\theta}(\mathbf{z}^s|\mathbf{z}^g)$.
* [uNLg, KD5e, tfzG]: we’ll add comparisons to two additional requested baselines (IODINE, beta-VAE).

We will post the results of these new experiments here as a comment once they are ready, and will also update the manuscript.

Based on the reviews, we observed that two important benefits of our method over prior works were unnoticed. Hence, we'll now explicitly emphasise them here and in the manuscript:

* We'll explicitly describe the advantages of our canonical object representation over recent works (spatial mixture models). This includes computational efficiency when rendering unbounded scenes, and not wasting the model's representation capacity to model each object at every possible position.
* We'll emphasise computational efficiency of the MCMC scheme on our structured graphical model as (1) it allows caching computation and only re-rendering parts of the scene that need to be considered for a proposed change (e.g. just background); (2) each MH step need not revert any progress made on other variables (e.g. if the background is perfectly inferred but objects are not yet, then an MH proposal improving an object will not change the background).

Based on the reviewers’ questions, we will make the following clarifications to the manuscript:


* [uNLg, tfzG, KD5e] We’ll add analysis of new ablation studies, clearly stating the benefits of each method’s design choice.
* [uNLg, tfzG, KD5e] We'll add analysis of results in the experiments section, focusing on different axes of out-of-distribution generalization and the performance of each baseline.
* [uNLg] To avoid possible misunderstanding, we'll now explicitly state in the text that our work does not use any labelled supervision and does not have any prior knowledge about object relationships; it instead learns these without supervision via the high-level prior $p_{\theta}(\mathbf{z}^s|\mathbf{z}^g)$.
* [gUkP, tfzG] We'll edit Equation 2 to clarify how our model is factored, including disentanglement of shape and appearance.

---

### Author Response · Authors · 2022-08-09
**Requested Experiments (1/3)**

In the following comment thread, we provide the requested ablation studies and additional baselines. Both demonstrate the points we argue in the paper. We summarize the results here; please see the updated manuscript for full results.

---

> ### Author Response · Authors · 2022-08-09
> **Requested Experiments (2/3)**
>
>
> # Ablation Study
>
> ### Ablating MCMC inference (amortised inference vs MCMC)
>
> Here we provide an ablation study on the effects of our novel MCMC scheme. We compare it with the standard approach of amortised inference (i.e. an encoder network predicts the posterior parameters). To keep this comment brief, we only give segmentation results on GQN dataset; see the updated paper for more tasks and datasets. The following table shows mean segmentation covering (mSC; higher is better).
>
> | GQN Mean Segmentation Covering            |Test (I.I.D.)                            |O.O.D.                        |
> |----------------|-------------------------------|-----------------------------
> |MCMC (ours) 		     |0.88            |$\mathbf{0.89}$
> |Encoder         |$\mathbf{0.91}$            |0.55
>
> These results confirm that amortised inference is a critical bottleneck for out-of-distribution generalization: though amortised inference performs well when the test distribution is identical to the training distribution (first column), its performance drops significantly on out-of-distribution images (second column), while MCMC holds up.
>
>
> ### Ablating generative model structure (MCMC on our model vs MCMC on unstructured generative model)
>
> Next we analyse the effects of using our compositional model compared to a non-compositional model, which has one latent variable rather than one per object (similar to NeRF-VAE), but otherwise with the same architecture as ours). Note the rest of the method is intact (e.g. we perform MCMC inference on both approaches). To keep this comment brief, we only show depth relative error (lower is better) on GQN; please see the paper for full results.
>
> | GQN Depth Relative Error         |Test (I.I.D.)                            |O.O.D.                        |
> |----------------|-------------------------------|-----------------------------|
> |MCMC on structured generative model (ours)  		     |0.031            |$\mathbf{0.034}$            |
> |MCMC on unstructured generative model         |0.031            |0.221            |
>
> These results demonstrate that our proposed compositional generative model significantly improves out-of-distribution generalization: though both models perform similarly on IID test data (first column), the unstructured model performs significantly worse on out-of-distribution images (second column).
>
>
> ### Ablating high-level prior over scene variables **$p_{\theta}(\mathbf{z}^s|\mathbf{z}^g)$**
>
> Finally analyse the effects of our high-level prior over scene variables $p_{\theta}(\mathbf{z}^s\mid\mathbf{z}^g)$.  Here, we evaluate samples generated by the model, comparing FID (lower is better) with ablated model, which samples $\mathbf{z}^s$ from the prior.
>
> |FID            |GQN                            |Arrow                        |
> |----------------|-------------------------------|-----------------------------|
> |Ours 		     |$\mathbf{80.3}$            |$\mathbf{141.4}$            |
> |Ours without  **$p_{\theta}(\mathbf{z}^s\mid\mathbf{z}^g)$**        |200.4            |275.7            |
>
> These results demonstrate that our hierarchical model with its high-level prior $p_{\theta}(\mathbf{z}^s\mid\mathbf{z}^g)$ is necessary to correctly model the density of scenes: our model samples plausible scenes as it can model relationships between objects, while the ablated model performs much worse.

---

> > ### Author Response · Authors · 2022-08-09
> > **Requested Experiments (3/3)**
> >
> > # Additional Baselines
> >
> >
> > Here we provide results from the additional baselines requested by reviewers. As with those already included in the paper, these baselines show a significant drop in performance on out-of-distribution images. Here we show only a subset of results for brevity; please see the updated paper and supplementary material for full results.
> >
> >
> > ### Slot-Attention with MCMC (requested by Reviewer uNLg)
> >
> > Here we adapt and apply our proposed MCMC inference to the learnt decoder of Slot-Attention (recall that we already provided results from Slot-Attention in the paper, but using their encoder). In this comment, we only show results on object segmentation, measuring mean segmentation covering (mSC; higher is better) -- please see the paper for more results. Note that Slot-Attention cannot perform 3D tasks, such as depth estimation.
> >
> > | GQN Mean Segmentation Covering         |Test (I.I.D.)                            |O.O.D.      |
> > |----------------|-------------------------------|-----------------------------|
> > |Ours         |$\mathbf{0.88}$            |$\mathbf{0.89}$            |
> > |Slot-Attention 		     |0.67            |0.56            |
> > |Slot-Attention with MCMC 		     |0.56            |0.54            |
> >
> > Comparing inference methods on Slot Attention, we see that MCMC under-performs relative to amortized inference on the IID test set (which is expected, since the encoder was trained on such data).
> > Slot-Attention with MCMC inference successfully reconstructs the input image during its test-time optimization, but this comes at the cost of lower segmentation performance. In contrast, our method performs well on both reconstruction and segmentation.
> > However, using MCMC results in a much smaller gap between IID and OOD settings, demonstrating the benefit of our proposal of using MCMC.
> > Still, both variants of Slot Attention perform significantly worse in both settings than our own method.
> >
> >
> > ### $\beta$-VAE (requested by Reviewer uNLg)
> >
> > Here we give results from $\beta$-VAE. This cannot perform any vision tasks apart from reconstruction, so we only measure reconstruction quality, using PSNR (higher is better).
> >
> > | GQN PSNR       |Test (I.I.D.)                            |O.O.D.                        |
> > |----------------|-------------------------------|-----------------------------|
> > | Ours		     |$\mathbf{24.1}$            |$\mathbf{21.8}$            |
> > | beta-VAE		     |20.6            |15.6            |
> >
> > We see significantly lower performance with $\beta$-VAE than our method, particularly in the OOD setting, even though $\beta$-VAE is intended to learn a disentangled latent space (which has been hypothesized to give greater robustness to distribution shifts).
> >
> >
> > ### IODINE (requested by Reviewer tfzG)
> >
> > Finally, we also provide results from another 2D baseline, IODINE, which uses iterative amortized inference. We tuned slot count, number of IAI iterations, pixel noise std, learning rate, and gradient clipping. For brevity we only show results for object segmentation, measuring mean segmentation covering (higher is better); please see the paper for more results. Note that IODINE cannot perform 3D tasks, such as depth estimation or NVS.
> >
> > | GQN Mean Segmentation Covering         |Test (I.I.D.)                            |O.O.D.                        |
> > |----------------|-------------------------------|-----------------------------|
> > |Ours         |$\mathbf{0.88}$            |$\mathbf{0.89}$
> > |IODINE 		     |0.54            |0.53           |
> >
> > IODINE with iterative amortized inference successfully reconstructs the input image during its test-time optimization, but this comes at the cost of lower segmentation performance. In contrast, our method performs well on both reconstruction and segmentation. We see that the iterative amortized inference scheme results in only a small drop in segmentation performance on OOD data compared with IID; however overall performance is substantially lower than with our model and its MCMC inference scheme (note that the original IODINE paper does not show successful results on textured data like our GQN dataset).

---

### Meta-Review · Area_Chair_qMqu · 2022-08-25

**Recommendation:** Accept
**Confidence:** Certain

**Metareview:**

This paper proposes a NERF-based object-centric VAE generative model, which it argues is a more "causal" generative model than prior attempts. While the approach is somewhat elaborate, it is described well, and involves a novel and quite well-motivated combination of several previously proposed components.

The MCMC sampler that improves the VAE encoder for out-of-distribution settings is an interesting additional contribution beyond the generative model, even though it is naturally computationally inefficient.

While the experimental settings are somewhat toy (all synthetic data), the proposed method is shown to enjoy substantial gains over some baselines representative of the closest related work, and the new baselines and ablation studies added during the response period are helpful.

After quite an engaging response period, the four thoughtful reviewers all agreed that the merits outweigh the shortcomings. I concur and recommend acceptance.

**Award:**

No

---

### Decision · Program_Chairs · 2022-09-14

Accept